# Structuring Semantic Embeddings for Principle Evaluation: A Kernel-Guided Contrastive Learning Approach

## Abstract

Evaluating principle adherence in high-dimensional text embeddings is challenging because principle-specific signals are often entangled with general semantic content. Our kernel-guided contrastive learning framework learns to disentangle these signals by projecting embeddings into a structured subspace. In this space, each principle is centered on a learnable **prototype kernel**—an optimized vector that embodies its core meaning—while a jointly learned **semantic basis** preserves context. A novel **offset penalty**, a loss term designed to create structure, then enforces a margin around each prototype. This ensures that even semantically similar principles are clearly separated while capturing their inherent contextual variability. Experiments show our optimized embeddings significantly improve performance on principle classification and ordinal regression, outperforming few-shot Large Language Models and demonstrating the value of specialized representations for reliable principle evaluation.

## 1 Introduction

Ensuring that the generated text adheres to human-defined principles—such as fairness, honesty, and safety—is a critical challenge for outputs from powerful language models (Weidinger et al., 2021; Bommasani et al., 2021; Hendrycks et al., 2023). Most research on AI alignment has focused on controlling models *during* the text generation process, aiming to make their outputs inherently safe or helpful (Christiano et al., 2017; Ouyang et al., 2022; Bai et al., 2022a). However, a separate and less-explored problem is how to reliably evaluate a text's adherence to principles *after* it has already been generated, a task known as post-hoc evaluation (Gehman et al., 2020; Rae et al., 2022). To tackle this challenge at scale, the dominant approach is to first encode the text into high-dimensional embeddings, which are designed to capture its rich semantic meaning, and then evaluate these embeddings against structured principle representations to determine whether the text adheres to the specified principles. The core difficulty, however, begins with these very embeddings. Standard general-purpose embeddings capture rich context and broad semantics, yet they are not explicitly structured or optimized to isolate the specific, often subtle, features indicative of principle adherence from general linguistic content (Devlin et al., 2019; Liu et al., 2022). This leads to embeddings where signals for nuanced principles, whose manifestations are frequently context-dependent (e.g., subtle bias, implied intent), are entangled with unrelated information (Bolukbasi et al., 2016; Caliskan et al., 2017). This entanglement poses a fundamental bottleneck to reliable post-hoc evaluation, with serious consequences in sensitive domains like automated content moderation and medical diagnosis (Birhane et al., 2021; Riegel et al., 2021; Esteva et al., 2021).

Addressing this representational bottleneck is therefore the logical first step towards bridging the evaluation gap. We therefore propose a novel **kernel-guided contrastive learning** framework that actively remodels the representation space. Specifically, our method first defines a learnable, central **prototype** for each principle, and then uses a novel **contrastive objective** to sculpt the space around these prototypes—pulling similar texts closer while enforcing a clear margin from dissimilar ones. The resulting structured embeddings can then be used as high-quality input features for any standard downstream classifier (e.g., an SVM or a simple regressor) to perform the final, reliable evaluation. To validate this two-stage approach, we test it on measurable proxy tasks—such as fine-grained

emotion and toxicity detection—that embody the same core challenge of disentangling subtle signals from a noisy background.

Our key contributions, aimed at enabling reliable principle alignment evaluation, are as follows: (1) **A novel conceptual framework** for principle alignment evaluation, utilizing prototype vectors as structured principle representations and a semantic basis to capture contextual dependencies. (2) **A neural principle architecture** that materializes this framework, using an attention mechanism to map embeddings to a structured subspace defined by learnable prototype kernels. (3) **A kernel-guided contrastive learning objective** featuring a novel offset penalty, specifically designed to organize this subspace by imposing a structured geometry around the prototypes for fine-grained principle separation. (4) **Extensive experimental validation** demonstrating the effectiveness of our optimized embeddings on various downstream principle evaluation tasks, including principle classification (GoEmotions), ordinal regression (Amazon Reviews), and classification in a sensitive domain (Toxic Comment Classification Challenge). Experiments show significant performance improvements over raw embeddings and superior results compared to few-shot Large Language Models in these specific evaluation contexts, validating our central thesis that optimizing representation geometry is a crucial and effective step towards reliable principle evaluation.

## 2 METHODOLOGY

Our framework is designed to restructure fixed, pre-computed text embeddings by projecting them into a principle-aligned subspace. This approach is distinct from end-to-end finetuning; instead of updating a large encoder's weights, we learn a separate, lightweight transformation that disentangles principle-specific features from general contextual information. The core idea is that learnable prototype kernels can serve as explicit anchors for principles within this subspace, much like basis vectors defining a coordinate system. This allows varied, context-dependent manifestations of a principle to be organized around a shared, abstract representation. Our framework realizes this by training a dedicated neural architecture with a novel, geometry-aware contrastive objective.

### 2.1 FRAMEWORK OVERVIEW

Our framework's central goal is to map a *fixed, high-dimensional* text embedding $\mathbf{X}_i \in \mathbb{R}^D$ into a low-dimensional, structured representation $\mathbf{e}_i \in \mathbb{R}^d$ (where $d \ll D$). This new representation is designed to make principle-specific features, which are entangled in the original space, readily discernible for downstream evaluators. Our framework consists of two core components:

**A Neural Architecture ($f_\theta$).**    We introduce a dedicated architecture, the *neural principle extractor*, which performs the mapping from $\mathbf{X}_i$ to $\mathbf{e}_i$. It uses an attention mechanism to project the input onto a subspace defined by a set of learnable prototypes $\{\mathbf{c}_k\}$, which represent the core meaning of each principle. The learnable nature of these prototypes is crucial, as their optimal positions are data-dependent (Details in Section 2.2).

**A Geometry-Aware Learning Objective ($\mathcal{L}_{total}$).**    This is a composite loss function that trains the neural architecture. It combines a supervised contrastive loss with our novel prototype offset penalty to explicitly organize the subspace, ensuring that the learned representations $\mathbf{e}_i$ are structured around their corresponding principle prototypes $\mathbf{c}_k$ (Details in Section 2.3).

### 2.2 NEURAL PRINCIPLE EXTRACTOR

The neural principle extractor, $f_\theta$, is designed to untangle principle-specific features from general context. It processes the input embedding $\mathbf{X}_i$ through two parallel streams that are ultimately fused.

The first stream produces the **Semantic Basis ($\mathbf{S}_i$)** by processing the input through a **shared Multi-Layer Perceptron (MLP)**. This vector's role is to capture the general topic or context of the text. The second stream produces the **Prototype Mapping ($\mathbf{m}_i$)** via an **attention mechanism**, which computes a weighted combination of a set of learnable **prototype kernels** $\{\mathbf{c}_k\}$. These kernels act as abstract, optimizable anchors for each principle in the learned subspace (initialization strategies are detailed in Appendix A.1).

For instance, given "This movie was a crushing disappointment," the Semantic Basis captures 'a movie review,' while the Prototype Mapping captures 'disappointment.' The final representation $\mathbf{e}_i$ is a weighted fusion of these two components, with their relative contribution controlled by a learnable parameter $\alpha$. This makes the final representation both principle-aware and contextually grounded. Further architectural details are available in Appendix A.2.

## 2.3 KERNEL-GUIDED CONTRASTIVE LEARNING

The neural principle extractor is trained by minimizing a composite loss function, $\mathcal{L}_{total}$, which we designed to progressively sculpt the geometry of the principle subspace. Our design philosophy is to combine standard learning procedures with novel, structure-enforcing objectives. We first use a **Contrastive Loss** to pull embeddings of the same principle into coarse clusters, and then introduce our novel **Offset Loss** to refine the local geometry around each prototype kernel by enforcing explicit separation margins. Finally, auxiliary terms including an **Orthogonality Loss** and a **Classification Loss** help ensure feature disentanglement and stabilize the training process. The total loss is a weighted combination of these components:

**Contrastive Loss ($\mathcal{L}_{\text{contrastive}}$).** This term encourages samples from the same principle to be closer in the learned embedding space while pushing apart samples from different principles. Based on the InfoNCE loss (Oord et al., 2018), our implementation incorporates adaptations like hard negative mining and dynamic class weights to enhance discrimination and handle class imbalance. The core form is:

$$\mathcal{L}_{\text{contrastive}} = \text{AdaptedInfoNCE}(\mathbf{e}_i, \{\mathbf{e}_k\}_{k \neq i}; \tau) \tag{1}$$

where $\mathbf{e}_i$'s are the enhanced features and $\tau$ is the temperature.

**Offset Loss ($\mathcal{L}_{\text{offset}}$).** While the contrastive loss encourages general class clustering, it does not explicitly control the shape of the clusters or the guaranteed separation between them. To impose a more precise geometric structure, we introduce the novel **Offset Loss**. This term acts as a geometric regularizer, controlling the position of a sample's kernel mapping $\mathbf{m}_i$ relative to the prototype kernels. It is composed of two penalties that work in tandem: an *Intra-Class Penalty* to allow for variation within a principle's cluster, and an *Inter-Class Penalty* to enforce separation between different principle clusters.

The Intra-Class Penalty introduces a "safe radius" ($\delta_{\text{intra}}$) around each prototype, only penalizing samples that fall outside this radius. This encourages compactness while preserving diversity. Here, $\mathbf{c}_{y_i}$ denotes the learnable prototype kernel corresponding to the true class label $y_i$. Its formulation is:

$$P_{\text{intra},i} = \max(0, ||\mathbf{m}_i - \mathbf{c}_{y_i}|| - \delta_{\text{intra}})^2 \tag{2}$$

The Inter-Class Penalty enforces a clear margin ($\delta_{\text{inter}}$) between clusters by ensuring that each sample is closer to its true prototype than to any incorrect one. Its formulation is:

$$P_{\text{inter},i} = \max(0, ||\mathbf{m}_i - \mathbf{c}_{y_i}|| - \min_{k \neq y_i} ||\mathbf{m}_i - \mathbf{c}_k|| + \delta_{\text{inter}})^2 \tag{3}$$

The total Offset Loss is a weighted sum of these two penalties:

$$\mathcal{L}_{\text{offset}} = \frac{1}{B} \sum_{i=1}^{B} w_{y_i} (\lambda_{\text{inclass}} P_{\text{intra},i} + \lambda_{\text{crossclass}} P_{\text{inter},i}) \tag{4}$$

where $B$ is the batch size, $w_{y_i}$ is the class weight for label $y_i$, and $\lambda_{\text{inclass}}, \lambda_{\text{crossclass}}$ are hyperparameters controlling the relative contributions of the intra-class and inter-class penalties.

**Orthogonality Loss ($\mathcal{L}_{\text{orthogonality}}$).** To encourage the semantic basis $\mathbf{s}_i$ and kernel mapping $\mathbf{m}_i$ to capture largely distinct information while allowing necessary interaction, this orthogonality loss promotes their 'soft' orthogonality. It is based on the cosine similarity between the two vectors, penalized when exceeding a dynamic margin $\delta_{\text{orthogonal}}$:

$$\mathcal{L}_{\text{orthogonality}} = \frac{1}{B} \sum_{i=1}^{B} w_{y_i} \cdot \max(0, |\cos(\mathbf{s}_i, \mathbf{m}_i)| - \delta_{\text{orthogonal}}) \tag{5}$$

where $w_{y_i}$ is a class weight.

**Classification Loss ($\mathcal{L}_{\text{classification}}$)** As an auxiliary objective, a standard classification loss is applied directly to the model's attention scores, which are interpreted as logits for class membership. This loss penalizes the model if it fails to assign a high score to the correct principle for a given input. Through backpropagation, this penalty signal directly trains the transformations that produce the query and key vectors, forcing the attention mechanism to learn the alignment between inputs and their corresponding principle prototypes. To mitigate class imbalance and focus on harder examples, we employ Focal Loss (Lin et al., 2017):

$$\mathcal{L}_{\text{classification}} = -\sum_i w_{y_i}(1 - p_{y_i})^\gamma \log p_{y_i} \tag{6}$$

where $p_{y_i}$ is the model's predicted probability for the true class $y_i$ (derived from softmax over the attention scores), $\gamma$ is the focusing parameter, and $w_{y_i}$ is a class weight.

**Magnitude Loss ($\mathcal{L}_{\text{magnitude}}$).** Specifically for ordinal regression tasks, this loss enforces the natural ordering of labels by encouraging the magnitude of the kernel mapping $||\mathbf{m}_i||$ to be proportional to the intensity of the ordinal label $I(y_i)$. This helps the learned subspace reflect the graded nature of ordinal values. The loss is:

$$\mathcal{L}_{\text{magnitude}} = \frac{1}{B} \sum_{i=1}^{B} (||\mathbf{m}_i|| - \lambda_{\text{mag\_scale}} I(y_i) \cdot ||\mathbf{c}_{y_i}||)^2 \tag{7}$$

where $\lambda_{\text{mag\_scale}}$ is a scaling factor and $I(y_i)$ maps the label to a numerical intensity (e.g., 1-5 for star ratings).

**Total Loss ($\mathcal{L}_{\text{total}}$).** The model is trained end-to-end by minimizing the total loss, a weighted sum of the above components:

$$\mathcal{L}_{\text{total}} = \lambda_{\text{contrastive}} \cdot \mathcal{L}_{\text{contrastive}} + \lambda_{\text{offset}} \cdot \mathcal{L}_{\text{offset}} + \lambda_{\text{class}} \cdot \mathcal{L}_{\text{classification}} + \lambda_{\text{orth}} \cdot \mathcal{L}_{\text{orthogonality}} (+\lambda_{\text{mag}} \cdot \mathcal{L}_{\text{magnitude}}) \tag{8}$$

The weights ($\lambda$ values, including $\lambda_{\text{mag}}$ for ordinal regression) are key hyperparameters determined through optimization, such as Bayesian optimization. A detailed analysis of the computational complexity can be found in Appendix A.4.

## 3 EXPERIMENT

This section evaluates the performance of our kernel-guided contrastive learning framework in enhancing principle evaluation in text embeddings. We detail our experimental setup in Section 3.1 and present results on three distinct datasets representing different principle evaluation tasks: GoEmotions, Amazon Reviews, and the Toxic Comment Classification Challenge. Our experiments demonstrate that embeddings optimized by our framework significantly improve downstream evaluation performance compared to using raw embeddings.

### 3.1 EXPERIMENTAL SETUP

All experiments utilize text embeddings (dimension $D = 1024$) generated by the `jina-embeddings-v3` (Jina AI, 2024) model, which has demonstrated strong performance in semantic similarity tasks relevant to principle alignment.

**GoEmotions Dataset.** GoEmotions (Demszky et al., 2020) is a large-scale corpus of Reddit comments annotated with 27 fine-grained emotion categories. For our experiments, we focus on a challenging subset of five principles (Disappointment, Sadness, Disapproval, Gratitude, Approval), including clusters of semantically similar emotions, to test the method's ability to distinguish subtle differences. This selection represents common positive and negative social emotions and allows rigorous evaluation of fine-grained discriminative capacity.

**Amazon Reviews Dataset.** The Amazon Reviews dataset (Ni et al., 2019) comprises user reviews and corresponding 1-5 star ratings for products on Amazon. These star ratings serve as indicators of sentiment intensity and represent ordinal values. The dataset provides a platform for validating our approach through sentiment classification (treating ratings as distinct classes) and ordinal regression

tasks (treating ratings as ordered values). A subset of this dataset was sampled and preprocessed, preserving the original, unbalanced distribution of star ratings.

**Toxic Comment Classification Challenge.** This dataset (Jigsaw & Kaggle, 2018) is critical for evaluating principle alignment in a sensitive domain – toxicity detection. It presents a highly unbalanced binary classification task (toxic vs. non-toxic), where we extract samples labeled 'non-toxic' (all toxicity labels are 0) and samples with only the 'toxic' label as 1. This forms an extremely unbalanced dataset (approximately 1:25 toxic vs. non-toxic labels in the test set, reflecting the real-world imbalance where toxic comments are much rarer) . The training set is balanced to approximately 1:3 using negative oversampling and positive undersampling. The task's difficulty is exacerbated by the less clearly defined nature of 'non-toxic' and 'toxic' principles.

Detailed data distributions are in Appendix C.3. Our principle extractor maps embeddings to $d = 64$ dimensions, chosen based on preliminary studies comparing different dimensions (Appendix A.5, showing $d = 64$ provides an optimal balance between performance and efficiency). Performance metrics are reported as Mean $\pm$ Standard Deviation over 10-fold cross-validation.

## 3.2 PERFORMANCE EVALUATION ON DOWNSTREAM TASKS

**GoEmotions Dataset: Classification.** Our evaluation on the GoEmotions dataset focuses on fine-grained principle classification. We train standard classifiers (SVM, Random Forest, Logistic Regression, XGBoost, Transformer) on both raw and optimized embeddings of the GoEmotions five-principle subset. Overall performance averaged over principles is summarized in Table 1. Detailed per-principle performance highlighting improvements on challenging principles is presented in Appendix B.1.

As summarized in Table 1, our optimized embeddings yield consistent and statistically significant improvements across all classifiers compared to raw embeddings. While XGBoost and Transformer also show improvements, the relative gains are most substantial for models that performed less strongly with raw embeddings, indicating our method particularly benefits classifiers struggling with the original representation. This highlights the effectiveness of our framework in creating a more discriminative representation space for principles, especially improving the performance floor.

Table 1: Overall (Avg. Principle) Performance on GoEmotions Five-Principle Set (Mean $\pm$ Std. Dev.)

| Metric | Emb. Type | SVM | RF | LR | XGBoost | Transformer |
|---|---|---|---|---|---|---|
| Precision | Raw Emb. | $0.748 \pm 0.049$ | $0.733 \pm 0.059$ | $0.737 \pm 0.031$ | $0.747 \pm 0.020$ | $0.785 \pm 0.031$ |
| | Opt. Emb. | $\mathbf{0.787} \pm 0.035$ | $\mathbf{0.789} \pm 0.036$ | $\mathbf{0.791} \pm 0.033$ | $\mathbf{0.773} \pm 0.028$ | $\mathbf{0.787} \pm 0.029$ |
| Recall | Raw Emb. | $0.721 \pm 0.045$ | $0.737 \pm 0.031$ | $0.722 \pm 0.031$ | $0.741 \pm 0.014$ | $0.763 \pm 0.024$ |
| | Opt. Emb. | $\mathbf{0.764} \pm 0.034$ | $\mathbf{0.769} \pm 0.039$ | $\mathbf{0.772} \pm 0.033$ | $\mathbf{0.765} \pm 0.039$ | $\mathbf{0.769} \pm 0.031$ |
| F1 | Raw Emb. | $0.729 \pm 0.046$ | $0.722 \pm 0.035$ | $0.726 \pm 0.031$ | $0.737 \pm 0.018$ | $0.764 \pm 0.036$ |
| | Opt. Emb. | $\mathbf{0.770} \pm 0.033$ | $\mathbf{0.767} \pm 0.036$ | $\mathbf{0.776} \pm 0.032$ | $\mathbf{0.764} \pm 0.026$ | $\mathbf{0.770} \pm 0.030$ |

**Amazon Reviews Dataset: Ordinal Regression.** On the Amazon Reviews dataset, we assess the utility of our optimized embeddings for principle evaluation, particularly in the context of ordinal tasks. The 1-5 star ratings in this dataset naturally represent ordered values, making it suitable for *ordinal regression* tasks, which capture the intensity of sentiment as an ordinal principle. We also evaluate performance on the associated *classification* tasks (treating ratings as distinct categories), with detailed results presented in Appendix B.3. Our framework is designed to enhance representations for both scenarios, but we focus the main text discussion on ordinal regression as it directly leverages the magnitude learning objective.

Table 2 summarizes overall ordinal regression performance across different regressors. Optimized embeddings consistently improve overall metrics (MSE, RMSE, $R^2$) compared to raw embeddings. Detailed per-class MSE, showing significant reductions for most star ratings, is provided in Appendix B.2. This demonstrates improved ability to capture the graded nuances of sentiment as an ordinal principle.

Table 2: Overall Ordinal Regression Performance on Amazon Reviews (Mean $\pm$ Std. Dev.)

| Metric | Emb. Type | SVM | RF | LR | XGBoost | Transformer |
|---|---|---|---|---|---|---|
| MSE | Raw Emb. | $0.668 \pm 0.135$ | $0.506 \pm 0.120$ | $0.635 \pm 0.097$ | $0.546 \pm 0.163$ | $0.602 \pm 0.173$ |
| | Opt. Emb. | $\textbf{0.365} \pm 0.158$ | $\textbf{0.392} \pm 0.143$ | $\textbf{0.394} \pm 0.149$ | $\textbf{0.377} \pm 0.097$ | $\textbf{0.359} \pm 0.086$ |
| RMSE | Raw Emb. | $0.813 \pm 0.083$ | $0.706 \pm 0.083$ | $0.795 \pm 0.059$ | $0.731 \pm 0.111$ | $0.768 \pm 0.110$ |
| | Opt. Emb. | $\textbf{0.593} \pm 0.119$ | $\textbf{0.617} \pm 0.107$ | $\textbf{0.618} \pm 0.112$ | $\textbf{0.609} \pm 0.080$ | $\textbf{0.595} \pm 0.071$ |
| $R^2$ | Raw Emb. | $0.604 \pm 0.086$ | $0.700 \pm 0.074$ | $0.624 \pm 0.060$ | $0.677 \pm 0.095$ | $0.643 \pm 0.103$ |
| | Opt. Emb. | $\textbf{0.785} \pm 0.089$ | $\textbf{0.770} \pm 0.080$ | $\textbf{0.768} \pm 0.083$ | $\textbf{0.777} \pm 0.055$ | $\textbf{0.788} \pm 0.047$ |

**Toxic Comment Classification Challenge.** For principle alignment evaluation in a sensitive domain, we assess our framework's performance on the Toxic Comment Classification Challenge dataset.

Table 3 summarizes the key results. Optimized embeddings consistently yield statistically significant improvements in Average F1 and Minority F1 across all classifiers compared to raw embeddings.

Table 3: Performance on Toxic Comment Classification Challenge (Mean $\pm$ Std. Dev.)

| Metric | Emb. Type | SVM | RF | LR | XGBoost | Transformer |
|---|---|---|---|---|---|---|
| Avg. F1 | Raw Emb. | $0.932 \pm 0.004$ | $0.949 \pm 0.003$ | $0.897 \pm 0.004$ | $0.918 \pm 0.004$ | $0.956 \pm 0.004$ |
| | Opt. Emb. | $\textbf{0.938} \pm 0.004$ | $\textbf{0.949} \pm 0.004$ | $\textbf{0.936} \pm 0.003$ | $\textbf{0.943} \pm 0.004$ | $\textbf{0.959} \pm 0.004$ |
| Minority F1 | Raw Emb. | $0.497 \pm 0.025$ | $0.405 \pm 0.044$ | $0.396 \pm 0.018$ | $0.433 \pm 0.024$ | $0.574 \pm 0.027$ |
| | Opt. Emb. | $\textbf{0.507} \pm 0.024$ | $\textbf{0.537} \pm 0.035$ | $\textbf{0.493} \pm 0.023$ | $\textbf{0.518} \pm 0.028$ | $\textbf{0.589} \pm 0.023$ |

## 3.3 COMPARISON WITH FEW-SHOT LARGE LANGUAGE MODELS

We compare our method's performance with that of few-shot prompted Large Language Models (LLMs), where the models directly perform the evaluation tasks based on prompting, serving as a generalist baseline. We evaluated `LLama3.3_70b_Q4_K` (Meta AI Team, 2025), and additionally tested `deepseek-chat-v3-0324` (DeepSeek Team, 2025) and `gemini-2.5-pro-exp-03-25` (Google DeepMind Team, 2025) via API calls.

Table 4 summarizes the comparison. Our method, using optimized embeddings with Transformer, consistently outperforms few-shot LLMs on these principle alignment tasks. This highlights the advantage of task-specific representation refinement for precise principle alignment compared to general-purpose LLM prompting. Detailed LLM evaluation results are in Appendix D.2.

Table 4: Performance Comparison with Few-shot Large Language Models

| Dataset | Metric | LLama3.3 | DeepSeek-chat-v3 | Gemini-2.5-pro | Opt. Emb. + Transformer |
|---|---|---|---|---|---|
| GoEmotions | F1-score | 0.67 | 0.70 | 0.70 | **0.77** |
| Amazon Reviews | MSE | 0.60 | 0.45 | 0.56 | **0.36** |
| Toxic Comment | Avg. F1 | 0.91 | 0.89 | 0.91 | **0.96** |

## 3.4 ABLATION STUDY

To understand the contribution of each component of our kernel-guided contrastive learning framework to the observed performance improvements, we conducted an ablation on the GoEmotions dataset study by selectively removing or isolating each loss term during training. Table 5 summarizes the performance across different configurations, showing F1 scores per principle and the average.

The results in Table 5 demonstrate that the combination of all three loss components yields the highest overall performance (0.78 average F1) and generally the best per-principle scores on this dataset. The **Classification Loss** is essential for basic discriminability, but insufficient on its own without

Table 5: Ablation study on the GoEmotions dataset (F1 score Mean). Disappt.-Disappointment, Sad.-Sadness, Disapprv.-Disapproval, Grat.-Gratitude, Apprv.-Approval.

| Configuration | Disappt. | Sad. | Disapprv. | Grat. | Apprv. | Average |
|---|---|---|---|---|---|---|
| Only Contrastive Loss | 0.37 | 0.75 | 0.72 | 0.92 | 0.74 | 0.75 |
| Only Offset Loss | 0.40 | 0.75 | 0.72 | 0.94 | 0.77 | 0.77 |
| Only Classification Loss | 0.26 | 0.58 | 0.61 | 0.85 | 0.61 | 0.64 |
| Without Contrastive Loss | 0.42 | 0.77 | 0.72 | 0.94 | 0.77 | 0.77 |
| Without Offset Loss | 0.44 | 0.71 | 0.73 | 0.93 | 0.76 | 0.77 |
| Without Classification Loss | 0.36 | 0.74 | 0.70 | 0.94 | 0.77 | 0.76 |
| Raw Embeddings | 0.36 | 0.64 | 0.66 | 0.93 | 0.72 | 0.72 |
| Optimized Embeddings (Full Model) | 0.49 | 0.77 | 0.72 | 0.94 | 0.76 | 0.78 |

explicit structure guidance. The **Contrastive Loss** and **Offset Losses** are crucial for improving the structural separation in the learned space, as shown by their better performance compared to using only classification loss. While removing either structural loss slightly reduces the average F1, their *combined effect* alongside the classification objective is necessary for optimal performance, particularly evident for the challenging 'Disappointment' principle where the full model achieves the highest F1 (0.49). Consistent with the main results, all configurations involving the learned losses significantly outperform using raw embeddings (0.72 on average), validating the overall approach to optimize the embedding space. This study, conducted on the GoEmotions dataset, demonstrates that integrating these complementary loss components is key to learning a balanced and robust principle-aware representation for effective evaluation.

## 3.5 EMBEDDING SPACE ANALYSIS

To understand the impact of our kernel-guided contrastive learning framework on the structure of the embedding space, we perform both qualitative visualization and quantitative geometric analysis.

**Qualitative Visualization using t-SNE.** We visualize the embedding spaces before and after optimization using t-SNE (Van der Maaten & Hinton, 2008) to qualitatively assess the separation of different principles or ratings.

Figure 1 compares the raw and optimized embedding spaces. Raw embeddings show significant overlap, especially for semantically similar principles or adjacent ratings. In contrast, optimized embeddings exhibit much clearer separability, forming distinct clusters for each principle/rating. For ordinal regression, the optimized clusters also show a clear ordered arrangement, consistent with the magnitude loss objective.

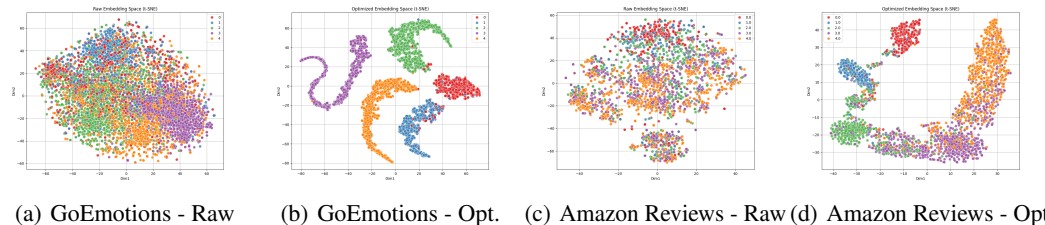

(a) GoEmotions - Raw  (b) GoEmotions - Opt.  (c) Amazon Reviews - Raw (d) Amazon Reviews - Opt.

Figure 1: Comparison of embedding spaces using t-SNE. (a) and (b) show raw and optimized embeddings for GoEmotions (classification). (c) and (d) show raw and optimized embeddings for Amazon Reviews (ordinal regression). Optimized embeddings demonstrate clearer separability and structure.

**Quantitative Geometric Analysis.** This qualitative observation of improved structure and separation in the embedding space is quantitatively supported by metrics analyzing cluster separability and quality. These metrics include the ratio between Within-class Variance and Between-class Variance

(Fisher, 1936), Silhouette Score (Rousseeuw, 1987), Class Overlap (Dom, 2001), and Fisher's Discriminant Ratio (Fisher, 1936). Table 6 summarizes the average results and standard deviations for each metric on the GoEmotions dataset embeddings.

Table 6: Quantitative Geometric Quality Metrics (Mean $\pm$ Std. Dev. )

| Metric | Emb. Type | Mean $\pm$ Std. Dev. | Interpretation | Improvement (%) |
|---|---|---|---|---|
| Within/Between Ratio | Raw | $8.76 \pm 0.53$ | Lower is better | 94.35% |
| | Optimized | $0.50 \pm 0.06$ | | |
| Silhouette Score | Raw | $0.018 \pm 0.004$ | Higher is better | 833.55% |
| | Optimized | $0.164 \pm 0.029$ | | |
| Class Overlap | Raw | $0.563 \pm 0.017$ | Lower is better | 34.12% |
| | Optimized | $0.371 \pm 0.038$ | | |
| Fisher's Discriminant Ratio | Raw | $0.165 \pm 0.008$ | Higher is better | 1008.87% |
| | Optimized | $1.825 \pm 0.236$ | | |

The results in Table 6 demonstrate a consistent and significant improvement across all measured geometric quality metrics after applying our kernel-guided contrastive learning optimization. This quantitative evidence strongly supports the visual observations from the t-SNE plots regarding enhanced class separability and improved cluster quality in the optimized embedding space. The improvements are statistically significant (as indicated by the separation of mean values relative to standard deviations), confirming that our optimization effectively structures the embedding space to enhance principle separability and cluster quality.

### 3.6 BEYOND PERFORMANCE: ADVANTAGES OF OPTIMIZED EMBEDDINGS FOR PRINCIPLE EVALUATION

Beyond the significant performance improvements on downstream evaluation tasks (3.2), our kernel-guided contrastive learning framework yields optimized text embeddings that offer several key advantages for principle evaluation. (1) It provides a *reusable intermediate representation* specifically structured for principle evaluation, enabling a modular pipeline across different tasks. This also facilitates *simplified downstream modeling*, allowing simple classifiers to achieve strong performance comparable to complex models on raw embeddings, significantly reducing model selection and tuning efforts. (2) Using these lower-dimensional embeddings enables *enhanced computational efficiency* for downstream evaluation. Training times for models like XGBoost were reduced by up to 96.5% compared to raw embeddings, making the evaluation process more practical. (3) The method yields a *structured subspace* for principle features (3.5), enhancing their discernibility and potential interpretability. These advantages highlight the utility of our approach for building robust and efficient principle evaluation systems beyond just metric gains.

## 4 RELATED WORK

**Principle Alignment in Text.** Prior efforts related to principle alignment often focus on constraining language models during text generation (e.g., RLHF (Christiano et al., 2017; Ouyang et al., 2022), DPO, Constitutional AI (Bai et al., 2022c)). Evaluating the principle alignment of already generated text or explicitly structuring embeddings for this evaluation purpose remains less explored. Our framework focuses on this evaluation gap by learning a discernible representation space.

**Semantic Contextualization and Subspace Learning.** Text embeddings capture context but may conflate subtle principle distinctions (Devlin et al., 2019; Liu et al., 2022). Subspace learning or task-specific projections aim to extract relevant features (Guo & Mackey, 2022; Wei et al., 2022; Gu & Roth, 2023). However, general-purpose methods like UMAP (McInnes et al., 2018) or Spectral Embedding (Von Luxburg, 2007), while useful for dimensionality reduction or structure visualization, are not optimized to specifically disentangle predefined principle-specific features. Our neural principle extractor learns a structured subspace specifically for principle evaluation, uniquely combining principle-specific extraction with a semantic basis via attention to preserve context dependence.

**Contrastive Learning.**   Contrastive learning enhances representation separability by pulling positives together and pushing negatives apart (Hadsell et al., 2006; Khosla et al., 2020). However, many supervised methods struggle with semantically similar principles because they contrast sample pairs, rather than explicitly modeling class structure. While Prototypical Contrastive Learning (PCL) (Li et al., 2021) addresses this by contrasting samples with prototypes computed from batch data, our approach differs by using learnable prototype kernels as explicit model parameters. We further impose a structured geometry around these kernels using a novel offset penalty for more fine-grained separation.

**Kernel-Based Methods and Prototype Learning.**   Kernel methods and prototype learning impose structure or learn representative points in embedding spaces (Schölkopf & Smola, 2002; Snell et al., 2017; Yang et al., 2016). Kernel offset constraints can refine separation (Wilson et al., 2017; Zhang et al., 2021). Our work integrates these ideas into a contrastive framework. Unlike methods that use prototypes for inference-time classification (Snell et al., 2017) or as batch-computed centers (Li et al., 2021), our learnable kernels serve as foundational parameters that actively define the target geometry of the representation space.

**Metric Learning.**   Metric learning aims to learn embedding spaces where distances reflect desired relationships (Yao et al., 2021). Objectives like triplet or N-pair loss are common, relying on relative distances between samples (Weinberger & Saul, 2009; Sohn, 2016). Our kernel-guided framework is a form of supervised deep metric learning, but differentiates itself by using explicit learnable prototype kernels as anchors and an offset penalty to structure the space around them, addressing the challenge of distinguishing semantically close principles beyond generic distance constraints based on sample pairs.

**Geometric Embeddings.**   Geometric embeddings represent structured data (e.g., knowledge graphs) as shapes for tasks like reasoning or querying (e.g., Query2Box (Ren et al., 2020) for KG querying). Our method similarly structures a representation space but applies learned point kernels to general text embeddings. Unlike geometric embeddings focusing on structured data relations, our objective is to enhance the distinguishability of principle-specific features in text via contrastive learning, using learned prototype kernels as anchors and attention to map text inputs to this space.

## 5   Conclusion and Future Work

This paper addresses the challenge of principle evaluation, a task hindered by the entangled, unstructured nature of standard text embeddings. We introduce a **kernel-guided contrastive learning** framework that tackles this representational bottleneck directly. Our method remodels the embedding space by learning a structured subspace where each principle is anchored by a learnable prototype kernel. A novel offset penalty then enforces a clear geometric separation between these principles, resulting in representations with significantly improved discernibility. Our experiments demonstrate that these structured embeddings boost the performance of various downstream evaluators and outperform few-shot LLMs. These results validate our central thesis: that explicitly remodeling the geometry of representation space is a critical and effective step towards building more reliable principle evaluation systems.

Current limitations include reliance on supervised data for known principles and unverified performance in diverse domains/languages. Application risks like misuse or bias amplification also warrant consideration. Future work targets evaluation with reduced supervision or for unseen principles, and extending the scope to diverse domains/languages. We will also investigate ethical considerations including misuse and bias. Technical extensions include continuous regression, automated annotation strategies, and dynamic environments. Integrating into evaluation components within RLAIF pipelines (Bai et al., 2022b; Lee et al., 2023) could enable more robust feedback signals.

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

## THE USE OF LARGE LANGUAGE MODELS (LLMS)

In preparing this work, we made limited and appropriate use of Large Language Models (LLMs) as follows:

- **Writing aid and polishing:** LLMs were used to assist in improving grammar, clarity, and style. The substantive content, ideas, and technical contributions remain the authors' own.
- **Retrieval and discovery:** LLMs were employed to support literature search and discovery (e.g., identifying related work). All cited references were verified by the authors.

# A IMPLEMENTATION DETAILS

This appendix provides further details regarding the neural principle extractor's architecture, prototype kernel initialization strategies, specific training hyperparameters and loss function configurations, computational complexity, and justification for the principle subspace dimension, as referenced in the main paper.

## A.1 PROTOTYPE KERNEL INITIALIZATION DETAILS

The $K$ learnable prototype kernels $\mathbf{c}_k \in \mathbb{R}^d$ are initialized based on the task type to encourage structured learning.

**For Classification Tasks:** For classification tasks (GoEmotions 5-principle set, Amazon Reviews classification), the $K$ prototype kernels are initialized randomly on the unit hypersphere in $\mathbb{R}^d$. To ensure distinct starting points and facilitate separation during training, we apply a procedure to guarantee a minimum pairwise Euclidean distance between any two initialized kernels. While not strictly enforcing orthogonality, this initial separation prevents kernels from collapsing onto the same point early in training. A target minimum distance of $\sqrt{2}$ (the Euclidean distance between orthogonal unit vectors) is aimed for during this initialization step.

**For Ordinal Regression Tasks:** For ordinal regression tasks (Amazon Reviews ordinal regression), the initialization incorporates the inherent order of the labels. The kernels are first initialized randomly on the unit hypersphere. Subsequently, the norm of the kernel $\mathbf{c}_k$ corresponding to ordinal label $k$ is scaled by a factor proportional to its numerical intensity $I(k)$. For the 1-5 star ratings in Amazon Reviews, $I(k) = k$. The scaling factor used is $(1.0 + (k - 1) \cdot \text{scale\_multiplier})$, where scale\_multiplier is a small constant (e.g., 0.1) to ensure that kernels corresponding to higher ratings have progressively larger initial magnitudes. This guides the magnitude loss and encourages the learned principle representations to exhibit an ordered structure.

## A.2 NEURAL NETWORK ARCHITECTURE DETAILS

The neural principle extractor $f_\theta$ is implemented as a neural network that maps the input text embedding $\mathbf{X}_i \in \mathbb{R}^{1024}$ to a $d = 64$ dimensional principle-aware representation $\mathbf{e}_i$. The architecture is composed of a shared Multi-Layer Perceptron (MLP) and an attention mechanism.

The shared MLP used to compute the semantic basis $\mathbf{s}_i$ consists of two fully connected layers with LeakyReLU activation functions and Batch Normalization. Dropout is applied after each hidden layer for regularization. The layer dimensions are as follows:

- Input layer: $\mathbb{R}^{1024} \to \mathbb{R}^{512}$
- Hidden layer 1: $\mathbb{R}^{512} \to \mathbb{R}^{256}$ (followed by LeakyReLU, Batch Norm, Dropout)
- Hidden layer 2: $\mathbb{R}^{256} \to \mathbb{R}^d$ (followed by LeakyReLU, Batch Norm, Dropout), where $d = 64$. The output of this layer is the semantic basis $\mathbf{s}_i$.

The Dropout rate used throughout the MLP is 0.2.

The attention mechanism involves linear transformations of the input embedding and the prototype kernels to compute queries, keys, and values:

- Query projection: query\_fc : $\mathbb{R}^{1024} \to \mathbb{R}^d$
- Key projection: key\_fc : $\mathbb{R}^d \to \mathbb{R}^d$
- Value projection: value\_fc : $\mathbb{R}^d \to \mathbb{R}^d$

These projected vectors are used in the scaled dot-product attention calculation as described in Section 2.3.

The learnable parameter $\alpha$ that weights the semantic basis and kernel mapping in the final fusion layer is a scalar variable initialized to 0.05.

### A.3 TRAINING AND LOSS FUNCTION DETAILS

This appendix provides detailed information regarding the training procedure and the full mathematical formulations and specific configurations for each component of the kernel-guided contrastive learning objective, as referenced from the main paper.

The neural principle extractor is trained end-to-end using the AdamW optimizer. The initial learning rate is set to 1e-4, with a weight decay of 1e-5. A learning rate scheduler, such as Cosine Annealing or ReduceLROnPlateau, is employed to dynamically adjust the learning rate during training based on validation performance. Training is performed for a maximum of 100 epochs, with early stopping based on performance on a validation set to prevent overfitting. The batch size used throughout our experiments is 128.

To handle class imbalance, which is particularly pronounced in datasets like Amazon Reviews and the Toxic Comment Classification Challenge, dynamic class weights $w_{y_i}$ are applied to the loss calculations. These weights are computed as the inverse frequency of each true class within the current training batch, providing stronger gradients for minority classes.

The training objective is to minimize a composite loss function $\mathcal{L}_{\text{total}}$, which is a weighted sum of several components (detailed below). The full mathematical formulations and specific parameters are provided for clarity and reproducibility.

**Contrastive Loss ($\mathcal{L}_{\text{contrastive}}$):** While the core idea is based on InfoNCE applied to enhanced features $\mathbf{e}_i$, our implementation incorporates specific adaptations for hard negative mining and handles class imbalance via dynamic weights $w_{y_i}$. The temperature parameter $\tau$ is set to 0.1. The specific adapted InfoNCE formulation used for $\mathcal{L}_{\text{contrastive}}$ is as follows, incorporating a term for hard negative samples:

$$\mathcal{L}_{\text{contrastive}} = \frac{1}{B} \sum_{i=1}^{B} w_{y_i} \left[ -\log \frac{\exp(\text{sim}(\mathbf{e}_i, \mathbf{e}_{p_i})/\tau)}{\sum_{k=1, k \neq i}^{B} \exp(\text{sim}(\mathbf{e}_i, \mathbf{e}_k)/\tau)} \right.$$
$$\left. -\lambda_{\text{hard}} \sum_{n \in \text{HardNegativeSet}(i)} \log(1 - \exp(\text{sim}(\mathbf{e}_i, \mathbf{e}_n)/\tau)) \right] \tag{9}$$

where $\mathbf{e}_{p_i}$ is a positive sample for $\mathbf{e}_i$ (another sample with the same true label in the batch), $\text{sim}(\cdot, \cdot)$ denotes cosine similarity, $\tau = 0.1$ is the temperature, $w_{y_i}$ is the dynamic class weight, HardNegativeSet($i$) is a subset of hard negative samples for $\mathbf{e}_i$ identified based on criteria like high attention scores towards $y_i$ or small embedding distance, and $\lambda_{\text{hard}}$ is a weighting factor for the hard negative term (tuned during optimization, typically between 0 and 1).

**Offset Loss ($\mathcal{L}_{\text{offset}}$):** This novel term regulates the position of a sample's kernel mapping $\mathbf{m}_i$ relative to its true principle prototype kernel $\mathbf{c}_{y_i}$ and other kernels, enforcing proximity to the correct kernel while maintaining distance from incorrect ones with controlled margins. The full loss definition is:

$$\mathcal{L}_{\text{offset}} = \frac{1}{B} \sum_{i=1}^{B} w_{y_i} (\lambda_{\text{inclass}} P_{\text{intra},i} + \lambda_{\text{crossclass}} P_{\text{inter},i}) \tag{10}$$

where $B$ is the batch size, $w_{y_i}$ is the dynamic weight, and $\lambda_{\text{inclass}}$, $\lambda_{\text{crossclass}}$ are hyperparameters (tuned during optimization). The penalty terms $P_{\text{intra},i}$ and $P_{\text{inter},i}$ are defined based on Euclidean distances to kernels and margins $\delta_{\text{intra}}$ and $\delta_{\text{inter}}$ (tuned during optimization, typically within [0.1, 0.5] for $\delta$ values):

$$P_{\text{intra},i} = \max(0, \|\mathbf{m}_i - \mathbf{c}_{y_i}\| - \delta_{\text{intra}})^2 \tag{11}$$

$$P_{\text{inter},i} = \max(0, \|\mathbf{m}_i - \mathbf{c}_{y_i}\| - \min_{k \neq y_i} \|\mathbf{m}_i - \mathbf{c}_k\| + \delta_{\text{inter}})^2 \tag{12}$$

**Orthogonality Loss ($\mathcal{L}_{\text{orthogonality}}$):** This term promotes "soft" orthogonality between the semantic basis $\mathbf{s}_i$ and kernel mapping $\mathbf{m}_i$ to encourage separation of information types. It is based on the

absolute cosine similarity, penalized when exceeding a dynamic margin $\delta_{orthogonal}$:

$$\mathcal{L}_{\text{orthogonality}} = \frac{1}{B} \sum_{i=1}^{B} w_{y_i} \cdot \max(0, |\cos(\mathbf{s}_i, \mathbf{m}_i)| - \delta_{orthogonal}) \tag{13}$$

where $w_{y_i}$ is a class weight. The dynamic margin $\delta_{orthogonal}$ is annealed linearly from an initial value of 0.5 down to a final value of 0.05 over the course of training epochs.

**Classification Loss ($\mathcal{L}_{\text{classification}}$):** A standard classification loss is applied using the attention scores as logits. To handle class imbalance and focus on harder examples, we employ Focal Loss (Lin et al., 2017). The full formula is:

$$\mathcal{L}_{\text{classification}} = - \sum_{i} w_{y_i} (1 - p_{y_i})^{\gamma} \log p_{y_i} \tag{14}$$

where $p_{y_i}$ represents the predicted probability for the true class $y_i$, $\gamma = 2$ is the focusing parameter, and $w_{y_i}$ is a class weight.

**Magnitude Loss ($\mathcal{L}_{\text{magnitude}}$):** Used specifically for ordinal regression tasks, this loss enforces the natural ordering by encouraging the magnitude of $\mathbf{m}_i$ to be proportional to the intensity of $I(y_i)$. The full formula is:

$$\mathcal{L}_{\text{magnitude}} = \frac{1}{B} \sum_{i=1}^{B} w_{y_i} (||\mathbf{m}_i|| - \lambda_{mag\_scale} I(y_i) \cdot ||\mathbf{c}_{y_i}||)^2 \tag{15}$$

where $w_{y_i}$ is a class weight, $\lambda_{mag\_scale}$ is a learnable scaling factor (initialized to 1.0), and $I(y_i)$ maps the ordinal label to a numerical intensity (e.g., $I(y_i) = y_i$ for 1-5 star ratings). This loss is applied only for ordinal regression tasks.

**Total Loss ($\mathcal{L}_{\text{total}}$):** The model is trained end-to-end by minimizing the total loss, which is a weighted combination of the above components:

$$\mathcal{L}_{\text{total}} = \lambda_{\text{contrastive}} \cdot \mathcal{L}_{\text{contrastive}} + \lambda_{\text{offset}} \cdot \mathcal{L}_{\text{offset}} + \lambda_{\text{class}} \cdot \mathcal{L}_{\text{classification}} + \lambda_{\text{orth}} \cdot \mathcal{L}_{\text{orthogonality}} (+ \lambda_{\text{mag}} \cdot \mathcal{L}_{\text{magnitude}}) \tag{16}$$

The weights ($\lambda$ values, including $\lambda_{\text{mag}}$ specifically for ordinal regression) are key hyperparameters that balance the contribution of each loss term. These weights, along with other hyperparameters like $\tau, \delta_{\text{intra}}, \delta_{\text{inter}}, \delta_{orthogonal}, \gamma, \lambda_{\text{hard}}$, and $\lambda_{mag\_scale}$ initialization, are determined through optimization, such as Bayesian optimization or extensive grid search on a validation set. Specific optimized lambda values used for each task/dataset are typically reported alongside the experimental results or in a dedicated section on hyperparameter tuning (e.g., Appendix C.2).

A.4 COMPUTATIONAL COMPLEXITY ANALYSIS

We analyze the computational complexity of our framework during training and inference.

**Training Complexity:** The primary computational cost during training arises from the forward and backward passes through the neural principle extractor and the calculation of the loss components over a batch of size $B$. The extractor involves:

- Shared MLP: A sequence of matrix multiplications. Given input dimension $D = 1024$, output dimension $d = 64$, and hidden dimensions $h_1 = 512, h_2 = 256$, the complexity is $O(D \cdot h_1 + h_1 \cdot h_2 + h_2 \cdot d)$ per sample.
- Attention Mechanism: Involves linear projections ($O(D \cdot d + d^2 \cdot K)$ for a batch of size $B$, where $K$ is the number of principles), computing attention scores ($O(B \cdot K \cdot d)$), and weighted summation ($O(B \cdot K \cdot d)$).

The dominant part of the forward pass per batch is approximately $O(B \cdot (D \cdot h_1 + h_1 \cdot h_2 + h_2 \cdot d + K \cdot d))$. Loss calculations involve vector operations and distance calculations on the $d$-dimensional embeddings and $K$ kernels:

- Contrastive Loss: $O(B^2 \cdot d)$ in the standard form, often optimized to $O(B^2)$ or $O(B \cdot P \cdot d)$ with $P$ positives per sample.

- Offset Loss: Involves distances to $K$ kernels, $O(B \cdot K \cdot d)$.
- Orthogonality, Classification, Magnitude Losses: $O(B \cdot d)$ or $O(B)$.

The overall training complexity per batch is dominated by the forward/backward passes and loss calculations, roughly $O(B \cdot (D \cdot h_{max} + K \cdot d) + B^2 \cdot d)$ in the worst case (for contrastive) or $O(B \cdot (D \cdot h_{max} + K \cdot d))$ with typical batch sizes and optimizations. This is comparable to other deep metric learning or contrastive learning frameworks.

**Inference Complexity:** Inference requires a single forward pass through the extractor. The complexity per sample is $O(D \cdot h_1 + h_1 \cdot h_2 + h_2 \cdot d + K \cdot d)$, which is linear with respect to $D$ and $K$. This makes obtaining the optimized embedding efficient.

**Downstream Efficiency Gains:** A practical benefit is the reduced computational cost for downstream tasks operating on the $d = 64$ dimensional embeddings compared to $D = 1024$ raw embeddings. This reduction is significant for many standard classifiers and contributes to faster downstream training and inference times. As noted in Section 3.6, this led to substantial training time reductions for downstream models.

### A.5 Justification for Principle Subspace Dimension ($d = 64$)

The choice of the principle subspace dimension $d = 64$ for the output embeddings was guided by preliminary experiments. We evaluated model performance on a validation set using various output dimensions (e.g., 32, 128, 256). $d = 64$ was found to provide a robust balance, offering significant dimensionality reduction from the input (1024 dimensions) while preserving sufficient information for effective principle discrimination in downstream tasks, yielding performance comparable to or better than higher dimensions with reduced computational cost for both model training and subsequent downstream task training/inference.

### A.6 Compute Resources

All experiments, including the training of the Neural Principle Extractor and evaluation of downstream models, were conducted on a machine equipped with four NVIDIA RTX 4090 GPUs (24GB VRAM each) and 128GB of system RAM. The CPU used was an Intel(R) Xeon(R) Platinum 8336C CPU @ 2.30GHz, running on Ubuntu 24.04 LTS.

Training of the Neural Principle Extractor is computationally efficient. A full training run typically completed within 3 to 15 minutes on a single NVIDIA RTX 4090, depending on the dataset size and complexity. Using multiple GPUs can further reduce this time. Inference using the trained extractor to produce optimized embeddings is significantly faster, requiring only a single forward pass per sample. Evaluating downstream models on the optimized embeddings is also substantially more efficient than using raw embeddings, as discussed in Section 3.6 and detailed in Appendix A.4.

## B Detailed Experimental Results

This appendix provides supplementary detailed results for the experiments presented in Section 3.

### B.1 GoEmotions Per-Principle Performance

This appendix provides detailed per-principle F1 performance for the GoEmotions dataset, complementing the overall results presented in Section 3.2. Table 7 shows the Mean $\pm$ Standard Deviation F1 scores for each of the five selected emotion principles.

The improvements are most pronounced for semantically similar and initially challenging principles with lower initial F1 scores, such as Disappointment and Sadness. Conversely, for principles like Gratitude, which already achieved high F1 scores with raw embeddings, the relative improvement is more modest across most classifiers. These results demonstrate that our method is particularly effective at refining distinctions for principles that are difficult to classify using standard embedding techniques, raising the performance ceiling for challenging cases while maintaining strong performance on easier ones.

Table 7: Per-Principle F1 Performance on GoEmotions Five-Principle Set (Mean ± Std. Dev.) in Appendix. Principles are abbreviated as Disappt., Sad., Disapprv., Grat., Apprv.

| Principle | Emb. Type | SVM | RF | LR | XGBoost | Transformer |
|---|---|---|---|---|---|---|
| Disappt. | Raw Emb. | 0.387 ± 0.090 | 0.237 ± 0.202 | 0.375 ± 0.054 | 0.359 ± 0.144 | 0.315 ± 0.073 |
| | Opt. Emb. | **0.482** ± 0.102 | **0.410** ± 0.087 | **0.479** ± 0.106 | **0.439** ± 0.099 | **0.386** ± 0.117 |
| Sad. | Raw Emb. | 0.643 ± 0.059 | 0.728 ± 0.069 | 0.672 ± 0.081 | 0.711 ± 0.082 | 0.711 ± 0.087 |
| | Opt. Emb. | **0.687** ± 0.048 | **0.734** ± 0.031 | **0.721** ± 0.054 | **0.698** ± 0.031 | **0.714** ± 0.034 |
| Disapprv. | Raw Emb. | 0.663 ± 0.074 | 0.691 ± 0.050 | 0.652 ± 0.070 | 0.703 ± 0.032 | 0.677 ± 0.055 |
| | Opt. Emb. | **0.720** ± 0.074 | **0.740** ± 0.064 | **0.728** ± 0.063 | **0.724** ± 0.082 | **0.733** ± 0.059 |
| Grat. | Raw Emb. | 0.925 ± 0.032 | 0.920 ± 0.024 | 0.921 ± 0.020 | 0.905 ± 0.031 | 0.915 ± 0.026 |
| | Opt. Emb. | **0.939** ± 0.021 | **0.940** ± 0.028 | **0.941** ± 0.024 | **0.934** ± 0.032 | **0.938** ± 0.029 |
| Apprv. | Raw Emb. | 0.732 ± 0.090 | 0.707 ± 0.031 | 0.727 ± 0.061 | 0.730 ± 0.060 | 0.716 ± 0.075 |
| | Opt. Emb. | **0.769** ± 0.053 | **0.747** ± 0.078 | **0.771** ± 0.055 | **0.762** ± 0.067 | **0.758** ± 0.053 |

## B.2    AMAZON REVIEWS PER-RATING PERFORMANCE

This appendix provides detailed per-rating performance for the Amazon Reviews dataset, supplementing the summarized classification and ordinal regression results presented in Section 3.2.

Table 11 shows the F1 performance for each star rating (1-5 S) on the Amazon Reviews dataset using raw and optimized embeddings.

Table 8: Classification F1 Performance per Rating on Amazon Reviews (Mean ± Std. Dev.) in Appendix

| Ratings | Emb. Type | SVM | RF | LR | XGBoost | Transformer |
|---|---|---|---|---|---|---|
| 1 - S | Raw Emb. | 0.712 ± 0.219 | 0.772 ± 0.166 | 0.713 ± 0.208 | 0.744 ± 0.235 | 0.731 ± 0.209 |
| | Opt. Emb. | **0.869** ± 0.112 | **0.874** ± 0.085 | **0.875** ± 0.084 | **0.894** ± 0.093 | **0.888** ± 0.090 |
| 2 - S | Raw Emb. | 0.277 ± 0.118 | 0.204 ± 0.213 | 0.297 ± 0.168 | 0.288 ± 0.221 | 0.432 ± 0.163 |
| | Opt. Emb. | **0.691** ± 0.187 | **0.708** ± 0.315 | **0.667** ± 0.176 | **0.760** ± 0.170 | **0.711** ± 0.141 |
| 3 - S | Raw Emb. | 0.433 ± 0.158 | 0.556 ± 0.176 | 0.503 ± 0.081 | 0.520 ± 0.141 | 0.584 ± 0.085 |
| | Opt. Emb. | **0.669** ± 0.106 | **0.697** ± 0.073 | **0.662** ± 0.112 | **0.657** ± 0.076 | **0.696** ± 0.143 |
| 4 - S | Raw Emb. | 0.478 ± 0.071 | 0.598 ± 0.114 | 0.565 ± 0.096 | 0.613 ± 0.078 | 0.558 ± 0.123 |
| | Opt. Emb. | **0.650** ± 0.094 | **0.639** ± 0.120 | **0.637** ± 0.129 | **0.622** ± 0.113 | **0.620** ± 0.105 |
| 5 - S | Raw Emb. | 0.614 ± 0.074 | 0.710 ± 0.099 | 0.676 ± 0.087 | 0.736 ± 0.069 | 0.724 ± 0.103 |
| | Opt. Emb. | **0.764** ± 0.112 | **0.766** ± 0.093 | **0.764** ± 0.115 | **0.741** ± 0.101 | **0.768** ± 0.082 |

Table 9 provides the per-rating Mean Squared Error (MSE) for the Amazon Reviews ordinal regression task.

## B.3    AMAZON REVIEWS CLASSIFICATION RESULTS

This appendix section provides detailed classification performance results on the Amazon Reviews dataset, supplementing the main text discussion which focuses on ordinal regression. For this task, the 1-5 star ratings are treated as distinct discrete categories.

Table 10 summarizes the overall (average per rating) classification performance across different classifiers using both raw and optimized embeddings.

Table 11 presents the F1 performance for each individual star rating (1-5) using both raw and optimized embeddings. Optimized embeddings generally show improved performance across most individual ratings, particularly for the intermediate ratings (2, 3, 4 stars) which are often more challenging to distinguish.

Table 9: Ordinal Regression MSE Performance per Rating on Amazon Reviews (Mean ± Std. Dev.) in Appendix

| Metric | Emb. Type | SVM | RF | LR | XGBoost | Transformer |
|--------|-----------|-----|-----|-----|---------|-------------|
| 1-S MSE | Raw Emb. | 0.483 ± 0.531 | 0.750 ± 1.207 | 0.567 ± 0.533 | 0.957 ± 1.145 | 0.350 ± 0.449 |
|         | Opt. Emb. | **0.177** ± 0.260 | **0.360** ± 0.543 | **0.140** ± 0.254 | **0.420** ± 0.555 | **0.157** ± 0.250 |
| 2-S MSE | Raw Emb. | 1.080 ± 0.867 | 1.415 ± 0.880 | 1.250 ± 0.972 | 1.465 ± 0.912 | 1.025 ± 1.072 |
|         | Opt. Emb. | **0.290** ± 0.386 | **0.435** ± 0.547 | **0.405** ± 0.334 | **0.335** ± 0.338 | **0.265** ± 0.363 |
| 3-S MSE | Raw Emb. | 0.726 ± 0.431 | 0.623 ± 0.235 | 0.804 ± 0.378 | 0.712 ± 0.269 | 0.539 ± 0.236 |
|         | Opt. Emb. | **0.386** ± 0.260 | **0.442** ± 0.312 | **0.442** ± 0.367 | **0.509** ± 0.361 | **0.376** ± 0.261 |
| 4-S MSE | Raw Emb. | 0.653 ± 0.221 | 0.320 ± 0.154 | 0.567 ± 0.211 | 0.340 ± 0.187 | 0.653 ± 0.260 |
|         | Opt. Emb. | **0.387** ± 0.171 | **0.433** ± 0.196 | **0.453** ± 0.195 | **0.380** ± 0.161 | **0.500** ± 0.189 |
| 5-S MSE | Raw Emb. | 0.607 ± 0.348 | 0.287 ± 0.149 | 0.467 ± 0.163 | 0.247 ± 0.095 | 0.567 ± 0.438 |
|         | Opt. Emb. | **0.427** ± 0.389 | **0.333** ± 0.365 | **0.400** ± 0.394 | **0.307** ± 0.200 | **0.313** ± 0.193 |

Table 10: Overall (Avg. Rating) Classification Performance on Amazon Reviews (Mean ± Std. Dev.)

| Metric | Emb. Type | SVM | RF | LR | XGBoost | Transformer |
|--------|-----------|-----|-----|-----|---------|-------------|
| Precision | Raw Emb. | 0.541 ± 0.038 | 0.627 ± 0.093 | 0.595 ± 0.058 | 0.630 ± 0.073 | 0.639 ± 0.067 |
|           | Opt. Emb. | **0.728** ± 0.077 | **0.726** ± 0.094 | **0.716** ± 0.084 | **0.718** ± 0.077 | **0.725** ± 0.062 |
| Recall | Raw Emb. | 0.522 ± 0.043 | 0.628 ± 0.083 | 0.583 ± 0.055 | 0.634 ± 0.060 | 0.628 ± 0.064 |
|        | Opt. Emb. | **0.721** ± 0.073 | **0.729** ± 0.080 | **0.715** ± 0.078 | **0.713** ± 0.069 | **0.723** ± 0.056 |
| Avg. F1 | Raw Emb. | 0.521 ± 0.041 | 0.609 ± 0.082 | 0.582 ± 0.052 | 0.619 ± 0.059 | 0.622 ± 0.061 |
|         | Opt. Emb. | **0.717** ± 0.074 | **0.721** ± 0.085 | **0.710** ± 0.081 | **0.708** ± 0.069 | **0.718** ± 0.058 |

## C  ADDITIONAL EXPERIMENTAL DETAILS

### C.1  DETAILS ON USED ASSETS AND LICENSES

This appendix provides details on the licenses and terms of use for the external datasets, embedding models, and language models used in this research, as referenced from the main paper. Our use of these assets adheres to their respective licenses and terms.

**Datasets.**

- **GoEmotions Dataset** (Demszky et al., 2020): This dataset is released under the **Creative Commons Attribution-ShareAlike 4.0 International License (CC BY-SA 4.0)**. Available at https://github.com/google-research/goemotions.

- **Amazon Reviews Dataset** (Ni et al., 2019): This dataset is provided for research purposes. Its use is subject to the terms specified by the data providers (e.g., Stanford/UCSD). Researchers should refer to the original source for specific usage guidelines. Available via the cited research project website.

- **Toxic Comment Classification Challenge**: This dataset, originally hosted on Kaggle (Jigsaw & Kaggle, 2018), is made available under the **CC0 1.0 Universal Public Domain Dedication**. Available at https://www.kaggle.com/c/jigsaw-toxic-comment-classification-challenge.

**Embedding Model.**

- **Jina Embeddings v3** (Jina AI, 2024): The embeddings used were generated by the jina-embeddings-v3 model. Jina AI models are typically licensed under the **Apache 2.0 License**. Researchers should consult the official Jina AI model documentation or Hugging Face model card for the most precise license information and terms of use.

Table 11: Classification F1 Performance per Rating on Amazon Reviews (Mean $\pm$ Std. Dev.) in Appendix

| Ratings | Emb. Type | SVM | RF | LR | XGBoost | Transformer |
|---|---|---|---|---|---|---|
| 1 - S | Raw Emb. | $0.712 \pm 0.219$ | $0.772 \pm 0.166$ | $0.713 \pm 0.208$ | $0.744 \pm 0.235$ | $0.731 \pm 0.209$ |
|  | Opt. Emb. | $\mathbf{0.869} \pm 0.112$ | $\mathbf{0.874} \pm 0.085$ | $\mathbf{0.875} \pm 0.084$ | $\mathbf{0.894} \pm 0.093$ | $\mathbf{0.888} \pm 0.090$ |
| 2 - S | Raw Emb. | $0.277 \pm 0.118$ | $0.204 \pm 0.213$ | $0.297 \pm 0.168$ | $0.288 \pm 0.221$ | $0.432 \pm 0.163$ |
|  | Opt. Emb. | $\mathbf{0.691} \pm 0.187$ | $\mathbf{0.708} \pm 0.315$ | $\mathbf{0.667} \pm 0.176$ | $\mathbf{0.760} \pm 0.170$ | $\mathbf{0.711} \pm 0.141$ |
| 3 - S | Raw Emb. | $0.433 \pm 0.158$ | $0.556 \pm 0.176$ | $0.503 \pm 0.081$ | $0.520 \pm 0.141$ | $0.584 \pm 0.085$ |
|  | Opt. Emb. | $\mathbf{0.669} \pm 0.106$ | $\mathbf{0.697} \pm 0.073$ | $\mathbf{0.662} \pm 0.112$ | $\mathbf{0.657} \pm 0.076$ | $\mathbf{0.696} \pm 0.143$ |
| 4 - S | Raw Emb. | $0.478 \pm 0.071$ | $0.598 \pm 0.114$ | $0.565 \pm 0.096$ | $0.613 \pm 0.078$ | $0.558 \pm 0.123$ |
|  | Opt. Emb. | $\mathbf{0.650} \pm 0.094$ | $\mathbf{0.639} \pm 0.120$ | $\mathbf{0.637} \pm 0.129$ | $\mathbf{0.622} \pm 0.113$ | $\mathbf{0.620} \pm 0.105$ |
| 5 - S | Raw Emb. | $0.614 \pm 0.074$ | $0.710 \pm 0.099$ | $0.676 \pm 0.087$ | $0.736 \pm 0.069$ | $0.724 \pm 0.103$ |
|  | Opt. Emb. | $\mathbf{0.764} \pm 0.112$ | $\mathbf{0.766} \pm 0.093$ | $\mathbf{0.764} \pm 0.115$ | $\mathbf{0.741} \pm 0.101$ | $\mathbf{0.768} \pm 0.082$ |

**Large Language Models (for Comparison).**

- **LLama 3.3** (Meta AI Team, 2025): The Llama 3 family of models is available under the **Llama 3 Community License**. Use of the quantized version (`LLama3.3_70b_Q4_K`) adheres to the terms of this license.
- **DeepSeek-Chat-v3** (DeepSeek Team, 2025): Used via API. Use is subject to **DeepSeek AI's API Terms of Service**.
- **Gemini 2.5 Pro** (Google DeepMind Team, 2025): Used via API. Use is subject to **Google's API Terms of Service** (e.g., Google AI or Google Cloud terms).

## C.2 HYPERPARAMETERS

## C.3 DATA DISTRIBUTION

Table 12: List of Hyperparameters

| # | Hyperparameter Name | Description | Current Value |
|---|---|---|---|
| | | **Neural Principle Extractor** | |
| 1 | input_dim | Dimension of the input feature vector. | (Variable) |
| 2 | num_classes | Number of classes in the classification task. | (Variable) |
| 3 | hidden_dims | Dimensions of the hidden layers in the shared MLP. | [512, 256] |
| 4 | output_dim | Dimension of the output feature vector from the extractor. | 64 |
| 5 | kernel_margin | Minimum distance between the initialized prototype kernels. | 1.414 |
| 6 | alpha | Weighting factor for combining semantic basis and principle-specific features. | 0.3 (trainable) |
| | | **Kernel-guided Contrastive Learning** | |
| 7 | temperature | Temperature coefficient for adjusting the similarity scaling. | 0.1 |
| 8 | class_weights | Weights for each class in the loss function. | (Computed by sklearn.utils.class_weight) |
| 9 | lambda_contrastive | Weight for the contrastive loss. | (Optimized using Bayesian optimization) |
| 10 | lambda_offset | Weight for the kernel offset loss. | (Optimized using Bayesian optimization) |
| 11 | lambda_classification | Weight for the classification loss. | (Optimized using Bayesian optimization) |
| 12 | lambda_orthogonality | Weight for the orthogonality loss. | (Optimized using Bayesian optimization) |
| 13 | lambda_magnitude | Weight for the magnitude loss. | (Optimized using Bayesian optimization) |
| 14 | offset_delta | Tolerance radius for in-class offset. | 0.63 |
| 15 | offset_margin | Additional penalty term for cross-class separation. | 2 |
| 16 | contrastive_k | Number of hard negative samples used in the contrastive loss. | 10 |
| 17 | focal_alpha | Weights for each class in the classification loss. | (Computed by class_weights) |
| 18 | focal_gamma | Controls the influence of easy and hard samples in the classification loss. | 2 |
| 19 | orthogonality_margin | Controls the strength of orthogonality in the optimization. | 0.5 |

Table 13: Data Distribution for GoEmotions, Amazon Reviews, and Toxic Comment Classification Challenge Datasets

| Dataset | Label | Train Count | Validation Count | Test Count |
|---|---|---|---|---|
| GoEmotions Dataset | Disappointment (0) | 709 | 91 | 88 |
| | Sadness (1) | 817 | 84 | 102 |
| | Disapproval (2) | 1200 | 212 | 195 |
| | Gratitude (3) | 1200 | 261 | 260 |
| | Approval (4) | 1200 | 258 | 236 |
| Amazon Reviews Dataset | 1 star (0) | 249 | 53 | 54 |
| | 2 stars (1) | 198 | 43 | 42 |
| | 3 stars (2) | 424 | 91 | 91 |
| | 4 stars (3) | 700 | 150 | 150 |
| | 5 stars (4) | 700 | 150 | 150 |
| Toxic Comments Dataset | Toxic (1) | 11064 | 580 | 870 |
| | Non-toxic (0) | 33192 | 14493 | 21739 |

## D EVALUATION LOG

### D.1 EMBEDDING LOG

**GoEmotions**

```
Model: SVM_RAW
   AUC: Mean=0.9198, Std=0.0172
```

```
   Precision: Mean=0.7483, Std=0.0485
   Recall: Mean=0.7207, Std=0.0454
   F1 Score (Minority Class 1): Mean=0.3867, Std=0.0899
   Overall F1 Score: Mean=0.7286, Std=0.0457
   F1 Score (Class 0): Mean=0.3867, Std=0.0899
   F1 Score (Class 1): Mean=0.6434, Std=0.0586
   F1 Score (Class 2): Mean=0.6626, Std=0.0738
   F1 Score (Class 3): Mean=0.9251, Std=0.0322
   F1 Score (Class 4): Mean=0.7315, Std=0.0899

Model: RF_RAW
   AUC: Mean=0.9064, Std=0.0260
   Precision: Mean=0.7334, Std=0.0592
   Recall: Mean=0.7366, Std=0.0305
   F1 Score (Minority Class 1): Mean=0.2372, Std=0.2015
   Overall F1 Score: Mean=0.7218, Std=0.0354
   F1 Score (Class 0): Mean=0.2372, Std=0.2015
   F1 Score (Class 1): Mean=0.7275, Std=0.0691
   F1 Score (Class 2): Mean=0.6905, Std=0.0498
   F1 Score (Class 3): Mean=0.9202, Std=0.0244
2025-05-09 11:58:19,124 - INFO -   F1 Score (Class 4): Mean
   =0.7067, Std=0.0311
2025-05-09 11:58:19,124 - INFO -
Model: LR_RAW
2025-05-09 11:58:19,124 - INFO -   AUC: Mean=0.9222, Std=0.0163
2025-05-09 11:58:19,124 - INFO -   Precision: Mean=0.7367, Std
   =0.0310
2025-05-09 11:58:19,124 - INFO -   Recall: Mean=0.7219, Std=0.0311
2025-05-09 11:58:19,124 - INFO -   F1 Score (Minority Class 1):
   Mean=0.3753, Std=0.0537
   Overall F1 Score: Mean=0.7261, Std=0.0306
   F1 Score (Class 0): Mean=0.3753, Std=0.0537
   F1 Score (Class 1): Mean=0.6715, Std=0.0811
   F1 Score (Class 2): Mean=0.6523, Std=0.0703
   F1 Score (Class 3): Mean=0.9214, Std=0.0201
   F1 Score (Class 4): Mean=0.7270, Std=0.0609

Model: XGB_RAW
   AUC: Mean=0.9266, Std=0.0204
   Precision: Mean=0.7470, Std=0.0203
   Recall: Mean=0.7412, Std=0.0144
   F1 Score (Minority Class 1): Mean=0.3591, Std=0.1436
   Overall F1 Score: Mean=0.7365, Std=0.0179
   F1 Score (Class 0): Mean=0.3591, Std=0.1436
   F1 Score (Class 1): Mean=0.7106, Std=0.0818
   F1 Score (Class 2): Mean=0.7034, Std=0.0315
   F1 Score (Class 3): Mean=0.9054, Std=0.0307
   F1 Score (Class 4): Mean=0.7298, Std=0.0600

Model: TRANSFORMER_RAW
   AUC: Mean=0.9344, Std=0.0146
   Precision: Mean=0.7715, Std=0.0358
   Recall: Mean=0.7639, Std=0.0347
   F1 Score (Minority Class 1): Mean=0.4150, Std=0.1180
   Overall F1 Score: Mean=0.7637, Std=0.0355
   F1 Score (Class 0): Mean=0.3154, Std=0.0728
   F1 Score (Class 1): Mean=0.7107, Std=0.0872
   F1 Score (Class 2): Mean=0.6774, Std=0.0545
   F1 Score (Class 3): Mean=0.9154, Std=0.0261
```

```
F1 Score (Class 4): Mean=0.7162, Std=0.0747

Model: SVM_OPT
   AUC: Mean=0.9360, Std=0.0143
   Precision: Mean=0.7868, Std=0.0350
   Recall: Mean=0.7639, Std=0.0343
   F1 Score (Minority Class 1): Mean=0.4818, Std=0.1023
   Overall F1 Score: Mean=0.7698, Std=0.0330
   F1 Score (Class 0): Mean=0.4818, Std=0.1023
   F1 Score (Class 1): Mean=0.6870, Std=0.0483
   F1 Score (Class 2): Mean=0.7200, Std=0.0737
   F1 Score (Class 3): Mean=0.9389, Std=0.0209
   F1 Score (Class 4): Mean=0.7689, Std=0.0531

Model: RF_OPT
   AUC: Mean=0.9279, Std=0.0154
   Precision: Mean=0.7758, Std=0.0345
   Recall: Mean=0.7685, Std=0.0385
   F1 Score (Minority Class 1): Mean=0.4104, Std=0.0866
   Overall F1 Score: Mean=0.7669, Std=0.0363
   F1 Score (Class 0): Mean=0.4104, Std=0.0866
   F1 Score (Class 1): Mean=0.7338, Std=0.0312
   F1 Score (Class 2): Mean=0.7403, Std=0.0635
   F1 Score (Class 3): Mean=0.9395, Std=0.0283
   F1 Score (Class 4): Mean=0.7468, Std=0.0775

Model: LR_OPT
   AUC: Mean=0.9359, Std=0.0126
   Precision: Mean=0.7913, Std=0.0334
   Recall: Mean=0.7718, Std=0.0328
   F1 Score (Minority Class 1): Mean=0.4794, Std=0.1059
   Overall F1 Score: Mean=0.7763, Std=0.0317
   F1 Score (Class 0): Mean=0.4794, Std=0.1059
   F1 Score (Class 1): Mean=0.7207, Std=0.0544
   F1 Score (Class 2): Mean=0.7279, Std=0.0634
   F1 Score (Class 3): Mean=0.9411, Std=0.0236
   F1 Score (Class 4): Mean=0.7707, Std=0.0547

Model: XGB_OPT
   AUC: Mean=0.9310, Std=0.0135
   Precision: Mean=0.7728, Std=0.0364
   Recall: Mean=0.7651, Std=0.0391
   F1 Score (Minority Class 1): Mean=0.4386, Std=0.0989
   Overall F1 Score: Mean=0.7643, Std=0.0358
   F1 Score (Class 0): Mean=0.4386, Std=0.0989
   F1 Score (Class 1): Mean=0.6978, Std=0.0313
   F1 Score (Class 2): Mean=0.7238, Std=0.0816
   F1 Score (Class 3): Mean=0.9339, Std=0.0315
   F1 Score (Class 4): Mean=0.7616, Std=0.0670

Model: TRANSFORMER_OPT
   AUC: Mean=0.9298, Std=0.0130
   Precision: Mean=0.7797, Std=0.0307
  Recall: Mean=0.7685, Std=0.0306
  F1 Score (Minority Class 1): Mean=0.3870, Std=0.0734
  Overall F1 Score: Mean=0.7701, Std=0.0302
  F1 Score (Class 0): Mean=0.3859, Std=0.1174
  F1 Score (Class 1): Mean=0.7137, Std=0.0339
  F1 Score (Class 2): Mean=0.7333, Std=0.0593
```

```
F1 Score (Class 3): Mean=0.9375, Std=0.0291
F1 Score (Class 4): Mean=0.7583, Std=0.0527

Comparison script finished.
```

**Amazon Reviews**

```
Model: SVM_RAW
   AUC: Mean=0.8540, Std=0.0200
   Precision: Mean=0.5409, Std=0.0383
   Recall: Mean=0.5215, Std=0.0434
   F1 Score (Minority Class 1): Mean=0.7120, Std=0.2189
   Overall F1 Score: Mean=0.5209, Std=0.0409
   F1 Score (Class 0): Mean=0.7120, Std=0.2189
   F1 Score (Class 1): Mean=0.2771, Std=0.1176
   F1 Score (Class 2): Mean=0.4332, Std=0.1575
   F1 Score (Class 3): Mean=0.4781, Std=0.0709
   F1 Score (Class 4): Mean=0.6141, Std=0.0736
   MSE: Mean=0.6675, Std=0.1352
   RMSE: Mean=0.8127, Std=0.0834
   R^2: Mean=0.6037, Std=0.0855
   MSE (Class 1-Star): Mean=0.4833, Std=0.5309
   MSE (Class 2-Star): Mean=1.0800, Std=0.8667
   MSE (Class 3-Star): Mean=0.7256, Std=0.4307
   MSE (Class 4-Star): Mean=0.6533, Std=0.2207
   MSE (Class 5-Star): Mean=0.6067, Std=0.3483

Model: RF_RAW
   AUC: Mean=0.8734, Std=0.0288
   Precision: Mean=0.6267, Std=0.0929
   Recall: Mean=0.6279, Std=0.0831
   F1 Score (Minority Class 1): Mean=0.7721, Std=0.1663
   Overall F1 Score: Mean=0.6091, Std=0.0820
   F1 Score (Class 0): Mean=0.7721, Std=0.1663
   F1 Score (Class 1): Mean=0.2038, Std=0.2129
   F1 Score (Class 2): Mean=0.5562, Std=0.1758
   F1 Score (Class 3): Mean=0.5983, Std=0.1135
   F1 Score (Class 4): Mean=0.7095, Std=0.0985
   MSE: Mean=0.5059, Std=0.1198
   RMSE: Mean=0.7064, Std=0.0831
   R^2: Mean=0.6998, Std=0.0739
   MSE (Class 1-Star): Mean=0.7500, Std=1.2071
   MSE (Class 2-Star): Mean=1.4150, Std=0.8804
   MSE (Class 3-Star): Mean=0.6233, Std=0.2354
   MSE (Class 4-Star): Mean=0.3200, Std=0.1543
   MSE (Class 5-Star): Mean=0.2867, Std=0.1492

Model: LR_RAW
   AUC: Mean=0.8498, Std=0.0205
   Precision: Mean=0.5952, Std=0.0577
   Recall: Mean=0.5830, Std=0.0550
   F1 Score (Minority Class 1): Mean=0.7133, Std=0.2077
   Overall F1 Score: Mean=0.5816, Std=0.0519
   F1 Score (Class 0): Mean=0.7133, Std=0.2077
   F1 Score (Class 1): Mean=0.2971, Std=0.1682
   F1 Score (Class 2): Mean=0.5032, Std=0.0808
   F1 Score (Class 3): Mean=0.5649, Std=0.0964
   F1 Score (Class 4): Mean=0.6762, Std=0.0871
   MSE: Mean=0.6349, Std=0.0965
```

```
RMSE: Mean=0.7946, Std=0.0592
R^2: Mean=0.6236, Std=0.0601
MSE (Class 1-Star): Mean=0.5667, Std=0.5325
MSE (Class 2-Star): Mean=1.2500, Std=0.9724
MSE (Class 3-Star): Mean=0.8044, Std=0.3784
MSE (Class 4-Star): Mean=0.5667, Std=0.2113
MSE (Class 5-Star): Mean=0.4667, Std=0.1633

Model: XGB_RAW
    AUC: Mean=0.8862, Std=0.0250
    Precision: Mean=0.6300, Std=0.0733
    Recall: Mean=0.6344, Std=0.0599
    F1 Score (Minority Class 1): Mean=0.7444, Std=0.2354
    Overall F1 Score: Mean=0.6192, Std=0.0591
    F1 Score (Class 0): Mean=0.7444, Std=0.2354
    F1 Score (Class 1): Mean=0.2876, Std=0.2212
    F1 Score (Class 2): Mean=0.5197, Std=0.1405
    F1 Score (Class 3): Mean=0.6128, Std=0.0784
    F1 Score (Class 4): Mean=0.7362, Std=0.0692
    MSE: Mean=0.5462, Std=0.1627
    RMSE: Mean=0.7307, Std=0.1110
    R^2: Mean=0.6768, Std=0.0949
    MSE (Class 1-Star): Mean=0.9567, Std=1.1450
    MSE (Class 2-Star): Mean=1.4650, Std=0.9116
    MSE (Class 3-Star): Mean=0.7122, Std=0.2693
    MSE (Class 4-Star): Mean=0.3400, Std=0.1873
    MSE (Class 5-Star): Mean=0.2467, Std=0.0945

Model: TRANSFORMER_RAW
    AUC: Mean=0.8916, Std=0.0340
    Precision: Mean=0.6389, Std=0.0667
    Recall: Mean=0.6279, Std=0.0640
    F1 Score (Minority Class 1): Mean=0.7306, Std=0.2088
    Overall F1 Score: Mean=0.6216, Std=0.0605
    F1 Score (Class 0): Mean=0.7306, Std=0.2088
    F1 Score (Class 1): Mean=0.4323, Std=0.1628
    F1 Score (Class 2): Mean=0.5835, Std=0.0853
    F1 Score (Class 3): Mean=0.5580, Std=0.1231
    F1 Score (Class 4): Mean=0.7237, Std=0.1026
    MSE: Mean=0.6020, Std=0.1734
    RMSE: Mean=0.7680, Std=0.1101
    R^2: Mean=0.6434, Std=0.1027
    MSE (Class 1-Star): Mean=0.3500, Std=0.4493
    MSE (Class 2-Star): Mean=1.0250, Std=1.0724
    MSE (Class 3-Star): Mean=0.5389, Std=0.2358
    MSE (Class 4-Star): Mean=0.6533, Std=0.2596
    MSE (Class 5-Star): Mean=0.5667, Std=0.4384

Model: SVM_OPT
    AUC: Mean=0.9268, Std=0.0203
    Precision: Mean=0.7281, Std=0.0773
    Recall: Mean=0.7208, Std=0.0731
    F1 Score (Minority Class 1): Mean=0.8693, Std=0.1118
    Overall F1 Score: Mean=0.7170, Std=0.0738
    F1 Score (Class 0): Mean=0.8693, Std=0.1118
    F1 Score (Class 1): Mean=0.6912, Std=0.1865
    F1 Score (Class 2): Mean=0.6690, Std=0.1056
    F1 Score (Class 3): Mean=0.6495, Std=0.0939
    F1 Score (Class 4): Mean=0.7642, Std=0.1115
```

```
MSE: Mean=0.3653, Std=0.1581
RMSE: Mean=0.5925, Std=0.1192
R^2: Mean=0.7847, Std=0.0885
MSE (Class 1-Star): Mean=0.1767, Std=0.2599
MSE (Class 2-Star): Mean=0.2900, Std=0.3859
MSE (Class 3-Star): Mean=0.3856, Std=0.2603
MSE (Class 4-Star): Mean=0.3867, Std=0.1707
MSE (Class 5-Star): Mean=0.4267, Std=0.3890

Model: RF_OPT
AUC: Mean=0.9314, Std=0.0239
Precision: Mean=0.7261, Std=0.0939
Recall: Mean=0.7290, Std=0.0797
F1 Score (Minority Class 1): Mean=0.8735, Std=0.0851
Overall F1 Score: Mean=0.7212, Std=0.0849
F1 Score (Class 0): Mean=0.8735, Std=0.0851
F1 Score (Class 1): Mean=0.7075, Std=0.3149
F1 Score (Class 2): Mean=0.6966, Std=0.0731
F1 Score (Class 3): Mean=0.6394, Std=0.1204
F1 Score (Class 4): Mean=0.7655, Std=0.0934
MSE: Mean=0.3915, Std=0.1433
RMSE: Mean=0.6165, Std=0.1073
R^2: Mean=0.7695, Std=0.0795
MSE (Class 1-Star): Mean=0.3600, Std=0.5426
MSE (Class 2-Star): Mean=0.4350, Std=0.5473
MSE (Class 3-Star): Mean=0.4422, Std=0.3120
MSE (Class 4-Star): Mean=0.4333, Std=0.1961
MSE (Class 5-Star): Mean=0.3333, Std=0.3651

Model: LR_OPT
AUC: Mean=0.9227, Std=0.0224
Precision: Mean=0.7160, Std=0.0842
Recall: Mean=0.7146, Std=0.0781
F1 Score (Minority Class 1): Mean=0.8752, Std=0.0844
Overall F1 Score: Mean=0.7099, Std=0.0813
F1 Score (Class 0): Mean=0.8752, Std=0.0844
F1 Score (Class 1): Mean=0.6672, Std=0.1763
F1 Score (Class 2): Mean=0.6619, Std=0.1115
F1 Score (Class 3): Mean=0.6366, Std=0.1294
F1 Score (Class 4): Mean=0.7643, Std=0.1152
MSE: Mean=0.3938, Std=0.1491
RMSE: Mean=0.6175, Std=0.1120
R^2: Mean=0.7684, Std=0.0827
MSE (Class 1-Star): Mean=0.1400, Std=0.2538
MSE (Class 2-Star): Mean=0.4050, Std=0.3343
MSE (Class 3-Star): Mean=0.4422, Std=0.3666
MSE (Class 4-Star): Mean=0.4533, Std=0.1950
MSE (Class 5-Star): Mean=0.4000, Std=0.3944

Model: XGB_OPT
AUC: Mean=0.9316, Std=0.0247
Precision: Mean=0.7184, Std=0.0771
Recall: Mean=0.7125, Std=0.0687
F1 Score (Minority Class 1): Mean=0.8944, Std=0.0927
Overall F1 Score: Mean=0.7078, Std=0.0688
F1 Score (Class 0): Mean=0.8944, Std=0.0927
F1 Score (Class 1): Mean=0.7598, Std=0.1698
F1 Score (Class 2): Mean=0.6569, Std=0.0760
F1 Score (Class 3): Mean=0.6215, Std=0.1125
```

```
F1 Score (Class 4): Mean=0.7411, Std=0.1006
MSE: Mean=0.3774, Std=0.0968
RMSE: Mean=0.6091, Std=0.0798
R^2: Mean=0.7768, Std=0.0552
MSE (Class 1-Star): Mean=0.4200, Std=0.5546
MSE (Class 2-Star): Mean=0.3350, Std=0.3377
MSE (Class 3-Star): Mean=0.5089, Std=0.3609
MSE (Class 4-Star): Mean=0.3800, Std=0.1607
MSE (Class 5-Star): Mean=0.3067, Std=0.2004

Model: TRANSFORMER_OPT
AUC: Mean=0.9191, Std=0.0259
Precision: Mean=0.7253, Std=0.0616
Recall: Mean=0.7227, Std=0.0564
F1 Score (Minority Class 1): Mean=0.8876, Std=0.0895
Overall F1 Score: Mean=0.7175, Std=0.0577
F1 Score (Class 0): Mean=0.8876, Std=0.0895
F1 Score (Class 1): Mean=0.7110, Std=0.1411
F1 Score (Class 2): Mean=0.6959, Std=0.1430
F1 Score (Class 3): Mean=0.6196, Std=0.1046
F1 Score (Class 4): Mean=0.7680, Std=0.0816
MSE: Mean=0.3592, Std=0.0856
RMSE: Mean=0.5951, Std=0.0714
R^2: Mean=0.7880, Std=0.0473
MSE (Class 1-Star): Mean=0.1567, Std=0.2495
MSE (Class 2-Star): Mean=0.2650, Std=0.3627
MSE (Class 3-Star): Mean=0.3756, Std=0.2606
MSE (Class 4-Star): Mean=0.5000, Std=0.1892
MSE (Class 5-Star): Mean=0.3133, Std=0.1933

Comparison script finished.
```

**Toxic**

```
Model: SVM_RAW
AUC: Mean=0.9577, Std=0.0081
F1 Score (Minority Class 1): Mean=0.4974, Std=0.0252
Overall F1 Score: Mean=0.9320, Std=0.0041

Model: SVM_OPT
AUC: Mean=0.8548, Std=0.0253
F1 Score (Minority Class 1): Mean=0.5069, Std=0.0241
Overall F1 Score: Mean=0.9375, Std=0.0036

Model: RF_RAW
AUC: Mean=0.9222, Std=0.0116
F1 Score (Minority Class 1): Mean=0.4051, Std=0.0440
Overall F1 Score: Mean=0.9487, Std=0.0032

Model: RF_OPT
AUC: Mean=0.9000, Std=0.0156
F1 Score (Minority Class 1): Mean=0.5370, Std=0.0350
Overall F1 Score: Mean=0.9488, Std=0.0041

Model: LR_RAW
AUC: Mean=0.9544, Std=0.0080
F1 Score (Minority Class 1): Mean=0.3958, Std=0.0177
Overall F1 Score: Mean=0.8969, Std=0.0044
```

```
Model: LR_OPT
    AUC: Mean=0.9430, Std=0.0122
    F1 Score (Minority Class 1): Mean=0.4931, Std=0.0233
    Overall F1 Score: Mean=0.9357, Std=0.0034

Model: XGB_RAW
    AUC: Mean=0.9374, Std=0.0102
    F1 Score (Minority Class 1): Mean=0.4326, Std=0.0239
    Overall F1 Score: Mean=0.9182, Std=0.0042

Model: XGB_OPT
    AUC: Mean=0.9442, Std=0.0095
    F1 Score (Minority Class 1): Mean=0.5184, Std=0.0280
    Overall F1 Score: Mean=0.9427, Std=0.0041

Model: TRANSFORMER_RAW
    AUC: Mean=0.9536, Std=0.0086
    F1 Score (Minority Class 1): Mean=0.5743, Std=0.0267
    Overall F1 Score: Mean=0.9561, Std=0.0040

Model: TRANSFORMER_OPT
    AUC: Mean=0.9508, Std=0.0112
    F1 Score (Minority Class 1): Mean=0.5885, Std=0.0229
    Overall F1 Score: Mean=0.9585, Std=0.0035

Cross-validation with all models completed.
```

## D.2 LLM Log

**LLM few shot prompting on GoEmotions**

```
You are an emotion classifier. I will provide you with a text, and
    you need to determine the main emotion expressed in the text
    and output the corresponding index of that emotion.

Here are some examples:

Text: "I hope Dallas gets close and loses in heartbreak."
Emotion: 0. disappointment

Text: "It could be worse. We could get [NAME] or that Philly
    traitor [NAME] back."
Emotion: 0. disappointment

Text: "To be fair, the world (especially politics) has been kind
    of a shit show since 2016"
Emotion: 0. disappointment

Text: "Don't let your high expectations of government disappoint
    you."
Emotion: 0. disappointment

Text: "I think it was, it was do scary, i honestly never wanna do
    that stuff again"
Emotion: 0. disappointment

Text: "You sound upset."
Emotion: 1. sadness
```

```
Text: "I'm so sorry. Read about you getting kicked out at home.
    That must be devastating"
Emotion: 1. sadness

Text: "I used to do the same exact thing! Now I love the fat on my
    steak and watching them cut it off in japanese restaurants
    makes me sad."
Emotion: 1. sadness

Text: "I miss my Shrek Ghrok :("
Emotion: 1. sadness

Text: "For some reason I wanted to be a bartender when I was 8.
    After hearing this story, makes me feel I missed out. "
Emotion: 1. sadness

Text: "it is very clearly incorrect, and it's apparent that your
    descent into reactionary politics has made you fully
    delusional. "
Emotion: 2. disapproval

Text: "This is why I could never work in construction"
Emotion: 2. disapproval

Text: "I don't like [NAME] but I'd never wish this fate to any
    parent."
Emotion: 2. disapproval

Text: "i say no or in da club if they come to damce with me i walk
    away"
Emotion: 2. disapproval

Text: "Not only that, the "improved controls" aren't a thing at
    all. I'm not re-buying Blood Money for nicer graphics. "
Emotion: 2. disapproval

Text: "Not gonna lie. Sucked out a few, but am really trying to
    analyze my play afterwards. Thanks!"
Emotion: 3. gratitude

Text: "ok thanks I'll give it a read and try to fact check"
Emotion: 3. gratitude

Text: "Thanks for the recommendations!"
Emotion: 3. gratitude

Text: "Nice, I didn't know that! Thanks for the info."
Emotion: 3. gratitude

Text: "Really glad you were there for her. I wish you both the
    best."
Emotion: 3. gratitude

Text: "Ah, fair enough."
Emotion: 4. approval

Text: "Needed that. A [NAME] ridiculous play is a game changer"
Emotion: 4. approval
```

```
Text: "it's horrid :/"
Emotion: 4. approval

Text: "I agree with this statement"
Emotion: 4. approval

Text: "Came on her face and she told you? Wow disrespect. Either
    she didn't know how you really felt or didn't care. Sorry
    about your luck"
Emotion: 4. approval

Emotion labels and their corresponding indices are as follows:

0. disappointment
1. sadness
2. disapproval
3. gratitude
4. approval

Please output only the index corresponding to the emotion, without
    any other content.

Here is the text to be classified:

I didn't know that, thank you for teaching me something today!
```

**LLM outputs on GoEmotions**

```
Processing text 1/881...
  Text: I'm really sorry about your situation :( Although I love
      the names Sapphira, Cirilla, and Scarlett!
  Original output: 1
  Predicted emotion index (0-4, -1 for invalid): 1
  Mapped predicted emotion index: 25
  Actual emotion index: 25
  Prediction correct

Processing text 2/881...
  Text: I didn't know that, thank you for teaching me something
      today!
  Original output: 3
  Predicted emotion index (0-4, -1 for invalid): 3
  Mapped predicted emotion index: 15
  Actual emotion index: 15
  Prediction correct

Processing text 3/881...
  Text: Thank you for asking questions and recognizing that there
      may be things that you don't know or understand about police
       tactics. Seriously. Thank you.
  Original output: 3
  Predicted emotion index (0-4, -1 for invalid): 3
  Mapped predicted emotion index: 15
  Actual emotion index: 15
  Prediction correct

Processing text 4/881...
  Text: You're welcome
```

```
Original output: 3
Predicted emotion index (0-4, -1 for invalid): 3
Mapped predicted emotion index: 15
Actual emotion index: 15
Prediction correct

Processing text 5/881...
  Text: 100%! Congrats on your job too!
  Original output: 4
  Predicted emotion index (0-4, -1 for invalid): 4
  Mapped predicted emotion index: 4
  Actual emotion index: 15
  Prediction incorrect

Processing text 6/881...
  Text: Girlfriend weak as well, that jump was pathetic.
  Original output: 2
  Predicted emotion index (0-4, -1 for invalid): 2
  Mapped predicted emotion index: 10
  Actual emotion index: 25
  Prediction incorrect

...........
```

**LLM few shot prompting on Amazon Reviews**

```
You are a product rating classifier. I will provide you with a
  customer review text, and you need to determine the product
  rating (number of stars) that the customer provided in the
  review, and output the corresponding integer from 1-5.

Here are some examples:

Text: "Lesson learned, next time I am going to research this a bit
    more and find foam that will actually LINE UP.  This foam is
    the worst for lining up the pieces to create a seamless studio
    look.  You can clearly see every cut out when hung up.  Will
    definitely not be buying again.  Next time I am going to look
    into pyramid foam, which is similar in price, and will
    actually look seamless by the time I am done hanging it on the
    wall.

Event when some of the egg foam pieces lined up between panels,
    they would eventually get off from one another  How is that
    even possible when they started off being lined up??!!??
    Bummer.

The description should state "foam will not line up when paneling
    together." Do not purchase if you want to think the foam will
    line up!"
Rating: 1.0

Text: "I am in pro audio & video for 30 years. I recently bought
    new Bose L1 Model II speakers and needed patch cables. The
    reviews on these are great so I bought 6 cables, some
    different lengths or ends. After 6 months I noticed one
    speaker was lower in volume and then eventually starting
    cutting off. I tested the cables and 3 of the 6 cables have a
    short between the RING & SLEEVE conductors. They tested fine
```

when I received them new. I am now replacing all 6 cables with
    another brand.
I liked the cables because they seemed sturdy and well made but I
    can no longer count on them in a professional environment.
If you own and of the Monoprice cables, I recommend that you test
    them with simple meter for shorts between the conductors.
I take very good care of my cables because I need them to work and
    I wish these would have worked but they have failed me. Keep
    testing these cables"
Rating: 1.0

Text: "When I pay top dollar for a Rode product I expect it to
    work in a professional environment and this product has been a
    disappointment. Yes, it does isolate the handling noise of a
    boom pole quite well but the rubber bands do not support the
    weight of my Rode NTG-3 mic. The mic sags in the bands and
    tends to fall out of the mount in use. Good thing the mic
    cable was still holding the mic or my mic would have crashed
    to the ground several times. Plus, you look like a fool when
    you flub the take because your mic slid out of the mount. I
    even wrapped a rubber band around the mic base to "catch" on
    the mount's rubber and the mic still slid out of the mount.
    This mount sort of works on a static boom but is pretty much
    worthless on a hand held boom pole. A nightmare in the field"
Rating: 1.0

Text: "Based on the positive reviews I ordered this, but had to
    return it right away.  I've owned another shure wireless mic
    system with a wireless beta 58 mic and had good experiences,
    but it was a different design than this.  I've also used other
    wireless mic systems from other manufacturers.  From the
    reviews I thought this would be a good economical way to get
    another shure system with two mic's.  When it arrived I hooked
    it up for a test and right away it was amplifying every
    single movement of my hand, and very loudly.  Maybe if I leave
    it in a mic stand I could do away with that, but this is a
    wireless mic so my intention is to be holding it so I can move
    around the stage or room.  Any slight movement of my hand on
    any part of the body of the mic is heard throughout the room.
    It's not acceptable. Picks up noise from handling mic body"
Rating: 1.0

Text: "I'm a long time musician, and a long time user of Ernie
    Ball Strings (on electric guitars). Unfortunately, I can't
    endorse these strings, even though they sound great and the
    light gauge saves my fingers. The problem is that they break,
    in my humble opinion, excessively easily. I put a set on and
    broke a g string within 2 days of moderate use. It snapped
    near the saddle, so I didn't think much of it. I replaced the
    set, and within a few more days I noticed that the wrapping on
    the g string was broken and becoming unraveled near my third
    fret. I bought 7 sets, so I still have 5 more sets to go
    through, but honestly I'll probably go back to elixirs when
    these are gone, since they seem to last longer.

*update*
I purchased these strings on 12/19. It' now 1/18, and I've broken
    3 strings from 3 different packs.
At this point, I really hate these strings. Break easily"

```
Rating: 1.0

Text: "My hopes for this pad was that it would be soft a sqishy on
    my shoulder.  The gel is pretty dense, so it's kind of hard
    and squishy and very heavy!  I found that it just added more
    weight.  I'm not using it.  I would look elswhere for a
    different product. Not as good as I expected"
Rating: 2.0

Text: "Works wonderfully. I don't think a snare stand will ever
    fit completely into any of the pouches if using the large one
    for the hihat stand, but having it stick out isn't a big deal.
     My one complaint is the idiotic placement of the shoulder
    strap loops - one is placed on the bottom and the other at the
     same spot on the top so that when lifting the bag from the
    strap it just rolls one way or the other along an axis through
     the center of the bag. It's very unstable and the handle in
    addition to the shoulder strap pretty much has to be used when
     carrying the bag.

Update after a year or two: this piece of gear is shredded on the
    inside of the pouches. It's extremely frustrating taking the
    gear in and out. The inside of the pouches is made of some
    sort of soft felt, which sounds great until you realize that
    there are no drum stands that have perfectly smooth shapes.
    Rubber feet, hat-stand spikes, pegs, screws, anything that
    sticks out can and will get caught on the felt and before long
     it turns into a web that instead of allowing your hardware to
     slide in and out makes it a struggle to pack and unpack. A
    struggle filled with cursing and people waiting to get on the
    dance floor you just played on because the DJ has set up
    already during the time it took you to wrestle all your stuff
    in to the shredded pouches.

Another thing I have found after many, many gigs and a few tours
    with this thing is that it is simply not conducive to setting
    up in a confined area. In order to get everything out you have
     to unroll it and take up a large footprint. So I end up
    taking up a ton of space while setting up whereas a regular
    bag or case would take up significantly less space. Very
    frustrating on a tiny stage or even on a big stage with a
    large band. Great piece of gear, could be better"
Rating: 2.0

Text: "Bought a brand new Behringer PMP1680S powered mixer and
    wanted a couple of new quality cables to go along with.
    Treated these cables like they were made of glass. In other
    word, very carefully. Hooked em up and I wasn't getting any
    sound out of the mains. After fooling around for 45 minutes
    thinking it was something I was doing wrong we gave up and
    just used the one channel. When I got home, I unscrewed the
    speakon side only to find that the wires literally fell out of
     the casing. I assume they were originally attached but the
    slightest tug must have made them fall out. Yea! China strikes
     again. I screwed it down and now it appears to be ok. I e-
    mailed Pyle Pro and it's been a week ago and ya know what they
     said.........Nothing. THEY NEVER RESPONDED. SPEAKON/SPEAKOFF
    ?"
Rating: 2.0
```

Text: "No instructions on using the nut slotting depth benchmark.
    All of the files seem to be about the same width, which makes
    the smaller string cuts way to wide.  The smaller strings
    slots need to be much narrower. Files too thick and no
    instructions"
Rating: 2.0

Text: "I will make this brief. I should have listened to other
    reviewers. I had to return it immediately because of the
    following. It has a major design flaw.

It is heavy and will not support itself unless the piece that
    attaches it to the mic stand has something immediately
    underneath it to support its weight, otherwise it will slide
    down and the screw will scratch your mic stand badly.

The part in my On Stage mic stand that had this criteria did not
    fit inside the quiklok piece which is made of metal so it can'
    t be modified in any way. So to me it was useless. Doesn't fit
     standard mic stands! (major design flaw)"
Rating: 2.0

Text: "We have home parties alot.....

I had a cheap 400 watter that did the trick using Froggy's Swamp
    Juice, but it eventually died.  No doubt from using too thick
    of fog juice.  BUT, I absolutely swear by Froggy's.  If your
    machine can't run it, get a better machine because Froggy's
    does an amazing job filling the area with fog.

Anyhow, this 1200 watter pushing Froggy's Backwood Bay juice cut
    at 2-3 parts juice to 1 part distilled water is really
    something else.  I hit the button once and the entire party
    area is filled - about 900 square feet.  I have to turn off
    the smoke detectors or it sets them off.  If I hit the button
    2 or 3 times within 30 minutes, you honestly cannot see 10
    feet in front of you.

I tend to hold the button down until it cycles off.  Wait about 15
     minutes and do it again, but note that the machine is ready
    again in about 5.  Then maybe once or twice more over the next
     couple of hours I cycle it again.  Then I unplug the machine
    and just let the fog hang in the air and eventually dissipate
    after maybe 4 hours from that first hit.

Put the right fog juice in THIS machine and it is the bomb!  Fills
     the room with fog that has great hang time and, cutting it
    with distilled water, it really is cost effective over the
    cheaper stuff.  The right machine and the right juice, cut
    down = 5 stars all the way!

Bad things (sorta):
1.  Had to move the fogger about 20 feet from the action as the
    fog shoots out really far with FORCE.
2.  Must turn off the breaker and unplug the smoke detectors.
    They will surely trip.

3.  Placement is a bit difficult given the size and weight of the
    machine, but especially the 20+ foot stream of forceful fog
    juice that comes shooting out the machine.

Lastly, I don't care about a timer.  Since I only need to hit the
    button 2 or 3 times total over a 4 hour period, a timer isn't
    warranted.

UPDATE AUGUST 2013:  HAD ONE DIE WITHIN A MONTH.  HAD TO DO A
    RETURN.  REPLACEMENT ONE GOING STRONG MONTHS LATER. WOW!
    Amazing fogger - QC issue"
Rating: 3.0

Text: "PROS:
* Decent quality strings that tune quickly and hold a tuning well
* Comfy to play on low EAD end-- which is how silk & steel is
    supposed to be
* Affordable price

CONS:
* Seems to my ear a bit more "tinny" in sound... not quite as deep
    , vibrant and mellow as standard strings.  This surprised me
    as these are supposed to have a "softer, more mellow" tone (
    which is why I bought these)
* The trebles of course seem no different than any treble of
    similar measure.
* Since silk & steel traditionally doesn't wear as long or as well
    as phosphor bronze, aside from the price I see no advantage
    to this type of string.

SPECIAL NOTE:  I like how the strings come two-to-a-paper.  The E
    is packed with g, A with b, D with e.  At first glance one
    thinks "that's cheap" however, it's smart marketing for
    several reasons:
1) It reduces cost of production so allows to keep retail costs as
    low as possible (it does cost something to make and print
    those sleeves)
2) It saves trees.  One would not believe how much wood is saved
    by simple, small conservation steps.  I remember a report from
    Celestial Tea which stated that annually they save something
    like 1 million trees by not putting paper tags on their
    teabags.  One never thinks about tiny things like that, but if
    we're going to save forests we need to reduce the use of
    paper however we can.

So 5 stars on conservation, but a "they're okay" on the strings
    themselves.  They're by no means a "bad" string; rather good
    in fact.  They just don't live up to the "silk & steel" claims
    I read before deciding to try out this type of string.  I'm
    probably going to be back with phosphor bronze before long.
    The low price however might seem attractive for students up
    front, but the low wear-time in the long run might not prove
    satisfactory, as from what I've heard these strings need to be
    replaced about twice as often as standard strings.

They're worth trying to see if you like the sound better.  These
    are just my personal observations.  Overall a good string;
    they just didn't strike me as "a reason to switch to silk &

steel" on a permanent basis Good strings but not a reason to
switch"
Rating: 3.0

Text: "Sounds ok but the position to rotate the dials is hard for
it to be mounted above dslr  camera. Plus xlr is not locking.
Meaning accidentally you can pull the cable off and you would
lose recording.

I also did a sound test.

Mics: Sennheiser MKE600 on line 1, Audio Technica AT8035 on line
2, both are set on mic stand and about 2.5' away from the
computer monitor that is playing a feature length film of mine
from 2004 that was ADRed. The setup is in our current decent
ADR room, so it's quiet enough.

Both Tascam and Zoom were providing phantom power to the mics

I was at the other room with the door closed, without doing much
of setting adjustments on either devices (assuming an indie
filmmaker like myself would not have time to go and carefully
set each as needed). I have only set recording to 48khz/24bits
.

On Tascam I have the gain set in mid and the dial at 3pm line (
both channels)
On Zoom, I have the dial set both on 5.1 (it's hard to get exact
as I was having seeing issue – old age).

Volume on both are set high for headphone.

so as you can see, the setting is not ideal but assuming that I'm
going to be running to set a shoot at a location and with the
chaos I may encounter when it's a one man crew or two men crew
 set...

btw, mounting the Zoom right above the camera is not a good thing
cause you can't see the dials or the screen (when tripod is
set at 5'6" tall). You will need a cage or additional hot
shoes support to mount the Zoom and the mic around the camera
(DSLR in this case), where as the Tascam can be mounted below
so with limited crew members, it is a fairly practical setup
than the Zoom. both limiters were set off and such.

I made sure the audio from the monitor is of the similar lines to
gage the recorded info. Tascam picks up better audio than the
Zoom did (maybe the settings were not correct). I do like the
packaging of the Zoom plus how it saves the audio into two
mono files unlike Tascam saving it the two tracks into a
stereo file with 2 channels.

Placing onto Premiere CS6, as it is, audio audible but quiet, not
hearing much of differences, then I added normalization to
both to -12db, Zoom sounded good but with a lot of noise... (
some friends of mine told me that the H4N had noise as well,
so not sure if this is the same issue or new problem).

Also: I did notice that for practicality wise, using the Tascam
    for a one man crew shoot would work best while the Zoom H6 is
    not feasible to access or read the screen when it is mounted
    above the camera. (Unless you use one of the magic arm).

Quick disclaimer: I'm not an audio engineering, but just an indie
    filmmaker that does a lot of stuff around Ohio and have
    learned to always have good audio with video (or worst case to
     do ADR... NOT). So I try to invest in decent mics and make
    sure get good audio that helps my visual creation... I did
    these tests on practicality and real-life environment since
    not every shoot will have a perfect sound, perfect room,
    perfect environment, etc... Look nice but not practical"
Rating: 3.0

Text: "I have to edit my once five star review because twice in a
    row I recieved a tuner in the mail, and after less than one
    week the screen goes totally dim. I Can't imagine this is the
    battery since that is supposed to last a year and I only used
    the tuner 4-5 times for less than a minute each time. It's so
    dim I can barley see the reading when I squint. It's only $10
    just a pain to wait two weeks for amazon to give a refund. I'
    ll be buying another one and trying it again! Good tuner,
    unlucky with two lemons I guess. Two were dead after 3 days,
    third times the charme!"
Rating: 3.0

Text: "I purchased this six-pack of colored cords several years
    ago. All of the cords worked fine when I purchased them, but
    over the years, I've had to resolder the connections on almost
     all of the cords. The problem is due to the conductors
    gradually sliding further and further out of the insulation
    until there's so much slack, they start coming into contact
    with each other at the pins and shorting out the audio signal.
     If you're handy with a soldering iron, a relatively easy
    resoldering job will fix the problem. I was able to extend the
     lifespan of all my cords.

I originally bought these colored mic cords for live use so that I
     could quickly trace a performer's mic to the mixer. They
    worked great for that, but they also posed a unique problem --
     the cords can really jump out in photographs! I never would
    have realized it, but then I saw pictures people had taken of
    our band. The multicolored spaghetti is a bit of an eyesore in
     my opinion, especially in flash photos against a dark stage.
    (FYI, I didn't dock the product any stars for this, because
    the cords are obviously intended to stand out.)

For the reason above, I went back to using all black cords on
    stage and moved these GLS into my recording studio. In the
    studio, multiple-colored mic cords are a real asset. In the
    years since, I've had to make a few additional repairs, but
    all six cords are still functional. Very handy, great for
    studio use, but longevity is questionable"
Rating: 3.0

Text: "i was initially curious to know whether these strings were
    really customized for flamenco or if it was all just a
    marketing gimmick of selling old wine in a new bottle. many

companies just package their products under a different name
or packaging to sell it as if its new. i just decided to go
for it & see for myself.
yes, its true that these strings are different from standard
daddario proarte. while proartes are delicate & rich in
harmonics, these strings are like a slap in the face, they've
a lot of bite to cut thru all kinds of noise.u don't hear any
sustaining harmonic trails when u strike a note, they just
strike really sharp. i changed from the proartes to the
flamenco set & there was this spike in the midrange & treble
frequencies. this helps the rhtyhm to really cut through.
especially my picado runs sound almost dirty, like someone
belting the guitar in a frenzy. the sustain is also a bit less
,so rasguedos don't get muddy, they get short & clear.
so its well suited for flamenco,dont buy it for any classical
playing. the black trebles also look kinda cool. price at $9
may be a bit steep, but if u r particular about having a sharp
 biting flamenco sound, this is it. flamenco strings for
flamenco players"
Rating: 4.0

Text: "This little ukulele does the job and doesn't have any
quality problems. It arrived only 3 days after I ordered it,
and in perfect shape.
The only annoying thing about the particular one I received is
that the volumne of the C-string dwarfs all of the other
strings. It's practically all you can hear when you strum (
with the open C) while tuned up to pitch. I'm guessing that's
just a fluke of this particular one, like a certain tone might
 resonate louder than any other in a particular room.
There are better-made ukuleles out there that come with gig-bags
and tuners for only 10-20 dollars more, but they don't have
the goofy pineapple print or cutaway in the headstock. :-)
Not knowing a thing about these prior to getting this one, I'd opt
 for a concert sized version next time. Good item for the
price"
Rating: 4.0

Text: "<a data-hook="product-link-linked" class="a-link-normal"
href="/Cordoba-22T-CE-Tenor-Cutaway-Ukulele/dp/B00JPN1XEK/ref=
cm_cr_arp_d_rvw_txt?ie=UTF8">Cordoba 22T-CE Tenor Cutaway
Ukulele</a>I gave this Crdoba 4 stars because the set up was
horrible.  Saddle as leaning forward so strings would not
intonate.  With some repair work and adjustments I was able to
 fix it and changing strings took care of the intonation
problem.  Now it plays in tune.  Euke has plenty of volume and
 sustain plugged in my acoustic amp or unplugged.  Workmanship
 is fine on the rosewood back and sides and solid spruce top.
 No fret buzzes.

I bought this instrument used from Amazon with the understanding
that it had no dings, scratches etc.  It arrived in a single
cardboard box with no padding and the top of the box was not
taped or secured in anyway.  Just amazing it arrived with only
 a small dent in the lower bout on the back.  Given the way it
 was packaged it could have been destroyed.  I thought about
sending it back but I like the instrument and can live with
the ding.  Crdoba makes fine instruments with supeb
workmanship.  They just need to do a better job with their set

```
     ups.  I am surprised Amazon would ship an instrument in a
     flimsy box with little support.  There was also no paperwork
     with the instrument. ... this Crdoba 4 stars because the set
     up was horrible. Saddle as leaning forward so strings would
     not ..."
Rating: 4.0

Text: "This is the most inexpensive keyboard stand that I have
     found. But for the price, you get an item that does what it
     should. It is obviously not meant to hold a heavy or large
     keyboard, but a 61 key instrument like the kind that Casio or
     Yamaha makes so many of will be fine with this.

If you need a more sturdy unit because you are using a keyboard
     with more weight to it, then really you should invest in a
     more sturdy and expensive model. I noticed that people are
     saying that the screws are not included. I at first thought
     that this was the case as well as they were not in the bag
     with the wrench and instructions. For the benefit of anyone
     who might be confused like I was, the allen wrench is strapped
      to the leg and the screws themselves are in the tops of the '
     X' section. You need to remove the screws and then put the
     legs on and mount the base legs. Why did they not put all the
     items in the plastic bag I do not know, but now you know where
      to look for them.

Inexpensive stand. Easy to set up. Works fine for light weight
     keyboards.
I am pleased with it. cheap stand that does the job"
Rating: 4.0

Text: "I have these in three guitars after trying out some
     undersaddle and sound hole pickups. Some peope miss the "quack
     " ,electric guitar type sound from these other types. I do not
     . I also like that the bridge and saddle do not have to be
     modified ,nor is any battery required. I run it through a
     Baggs Para-acoustic preamp or my Fishman Aura.
It really has to be installed by a professional repairman to be
     sure the sensors are located correctly.
Product was delivered safely and promptly. Most Natural Sound for
     the Money"
Rating: 4.0

Text: "Where do I begin???  For starters, I have been playing for
     almost 20 years and have been doing repairs on solid body
     electrics and acoustic guitars for 5 years now commercially.
     Not to sound arrogant - but I KNOW WHAT I'M TALKING ABOUT
     PEOPLE....

The craftsmanship for this guitar is excellent.  The construction
     is solid feeling and the paint finish is handsomely done.  The
      bridge is glued straight and even onto the body with no
     exposed seams. The nut is precisely cut for the strings - they
      sit there nice and snug.  On cheaply done 12 strings, you see
      that the nut is not cut very evenly.  This guitar feature a
     NUBONE brand nut & compensated saddle & that makes for great
     intonation on this guitar and your strings will stay in tune
     much longer.
```

The tuners are standard sealed tuners and do well holding the
    guitar in tune.  I normally like to replace the tuners
    immediately on my acoustics with "GROVER" brand tuners but
    after playing this guitar for a week straight – no less than 3
     hours a day (seriously folks), I've only had to tune the
    guitar twice.

The guitar arrived with the action set perfectly.  The strings
    were nice and low from the start and still had enough
    clearance for hard strumming with NO BUZZING.  The strings
    didn't need changing either since the guitar has D'Addario XL
    strapped on – great bright/jangly sounding highs with nice
    clear lows!!!

The body has a spruce top with mahogany sides & back. This guitar
    sounds great.  I compared it to my friends 18 year old FENDER
    brand 12 string and it sounds better than hers!  (I told her
    it sounds just as good as her guitar so as not to hurt her
    feelings. But in reality this guitar beats hers....LOL)

The electronics are not top of the line – obviously. BUT... The
    wiring is neat and the sound when amplified is pretty darn
    good.  Naturally, you need to reduce how much gain you use if
    you play really loud on your amp.  But that's just the way it
    is when dealing with acoustic guitars anyway.  I might in the
    future change it out for a PIEZO system but then again – I
    might not.  It holds up well as is.  The instrument control
    board also includes a nice handy tuner which always helps & it
     does the job accurately.
Kudos to the manufacturer for including an output jack for direct
    line connection to a soundboard/mixer besides the standard
    output jack you would plug into your amp.  That's real
    versatility and very helpful.  You usually don't find such
    versatility except on more expensive instruments!!!  Very nice
    , indeed.

  On the control panel for the electronics, the buttons are a
      little stiff when you press them. The knobs are a little too
       small to grab and a little too tight when you turn them.
      You have to have a moderately "soft touch" when pushing the
      buttons and a little more strength when using the knobs.
      Personally speaking, I tend to leave the "tone tweaking"
      alone on the actual guitars I own and always use my amp or
      pedals for tone shaping alteration anyway. So i won't be
      using the buttons or knobs too much anyway.

Lastly, I wish it came with a pick guard.  Oh well, I just ordered
     a nice one with a fancy hummingbird design on it.
Problem solved.

What's the bottom line, people???

BUY THIS GUITAR !!!!!!!!!

You'd be a fool not to.  You get so much for under $200.  What the
     heck are u waiting for???
I'm seriously gonna get a second one. GREAT INSTRUMENT FOR THE
    PRICE !!!"
Rating: 5.0

Text: "My application of this is to plug three guitars in to my
    pedalboard (see pics).  It is noiseless, and shows no tone-
    sucking qualities at all.  It runs well off a 9-volt or a
    Dunlop DC Brick, although that is just for the function of the
     lights.  It also works with no power at all (but the lights
    come in handy to know which selection is ON). As good as I
    hoped!"
Rating: 5.0

Text: "Being the owner of a couple different Heil microphones (the
     Heil Goldline Pro and the Heil Heritage) which never fail to
    impress, I decided it was time to step up to the Heil PR-40 to
     see what it was capable of as well. I purchased the Heil PR
    -40 to serve a few different purposes, mainly as a new mic for
     my ham radio, but also for use in VOIP applications, and
    finally to use for recording narration for videos. It suits
    all of these purposes just fine, but of course I knew this
    already since the same source is used for all those
    applications, that being my voice.

The mic itself is everything I would expect of a Heil microphone.
    It build ruggedly, and has a beautiful, flawless finish. The
    PR-40 comes in a padded leatherette carrying case that also
    holds the included mic clamp. The clamp has an adapter that
    screws in to allow the clamp to be used on different sized
    stands and boom arms. Also included was a Heil Sound decal. I
    have to say I'm disappointed the Heil mics don't come in the
    wooden presentation boxes they used to, but I am quite happy
    with the padded case they use now too.

The XLR jack on the mic is a little tight, it took a lot of effort
     to get the Neutrik XLR connector on my mic cable to lock into
     place. Removing the o-ring from the Neutrik connector on the
    mic cable allows the cable to lock into the PR-40 with no
    effort, but I prefer the slight amount of compression the o-
    ring provides to help keep dust out as well as prevent any
    rattling.

The thing about Heil mics is they pretty much occupy their own
    audio space in the world of mics. Nothing else comes close to
    sounding like a Heil mic, their timbre is unmistakable, but in
     a good way. A lot of people have trouble getting used to Heil
     mics because they're used to older design mics that need a
    lot of EQ to make them sound good. With Heil mics, they don't
    require an obscene amount of EQ to make them sound great, they
     pretty much sound great with the EQ set flat.

What I really like about the Heil PR-40 is it has a slightly
    scooped mid-range that takes the nasal "honk" and stuffiness
    out of my voice. I've never used a mic that sounded so
    broadcast-ready right out of box, and much of that is helped
    by the extended low-frequency response (when compared to most
    other dynamic mics) that picks up more of the deep chest
    resonance of the person talking or singing into the mic. Of
    course, this also makes the mic more prone to picking up low
    frequency rumble, but installing the mic in the optional
    accessory shock mount takes care of that problem without
    having to EQ out the low frequencies, which EQ really should

always be left as a last-ditch bandaid only anyways, as you're
    essentially EQ'ing out an entire octave of the human voice.

Proximity effect is really sweet on this mic. With the PR-40
    anything more than 6 inches away from the mouth starts
    sounding way too thin, with the sweet spot being around 2-4
    inches from the grill. However, you can literally get your
    lips right up against the grill and the proximity effect is
    still controlled to the point that the bass never becomes too
    thick or muddy, the mic just takes on an extremely warm,
    intimate sound that is absolutely spectacular.

Even though the mic comes with an internal sorbothane shock
    mounted capsule, the mic is still fairly prone to picking up
    low frequency through the mic stand or boom arm. I highly
    recommend getting the accessory shock mount to go with this
    mic, I guarantee it will make a world of difference.

What can I say? I'm more impressed by the mic than I thought I
    would be. I can gladly say I have absolutely no buyer's
    remorse whatsoever. This one is definitely a keeper.
    Remarkable Quality & Performance"
Rating: 5.0

Text: "This is the best guitar humidifier on the market. What
    makes it the best?

1. The synthetic sponge that holds so much more water then a
    typical sponge and it does NOT drip.
2. Single one time purchase with nothing else to buy (like humidi
    paks) The Best!"
Rating: 5.0

Text: "I've been playing the harmonica for over 43 years and this
    harmonica, made in Germany, exceeds my expectations.  I can
    bend single notes for melodic playing and with the usual
    vibrato over a wide variety of musical genres. Many music
    stores sell this same harmonica for $20 more than what I paid
    for it on Amazon. It pays to comparative shop. I've been
    playing the harmonica for over 43 years and ..."
Rating: 5.0

Please output only the integer from 1-5 corresponding to the
    rating, without any other content.

Here is the text to be classified:

I use several Behringer products (amps & pedals). I got this one
    last week and really hated the sound. The OD was WAY too harsh
    starting at level 1!

Lots of settings and tones, though. I changed out the tube for an
    Electro Harmonix 12AY7 and it sounds MUCH better. Don't expect
    "true" tube sound. It is, after all, only 1 preamp tube. It
    does give a "tube-like" OD sound, though. Close, but not exact
    . Good enough and built solid.

I was all ready to ship it back, but I'm now keeping it. Changed
    the tube, now I like it.}

**LLM outputs on Amazon Reviews**

```
Processing text 1/500...
  Text: My earlier review was for the Jr2- I don't know why it was
      posted for the Jr1! Well, anyway this is a nice guitar for
      the money. There is some initial buzzing in the beginning-
      but not anymore. It has great tone- it's clear and bright. I
       like that. Great sustain. Great travel guitar for adults.
      comes w/ a gig bag. Jr1 is a great guitar- buy one NOW! Good
      ! For Adults and travel. Jr1
  Original output: 5
  Predicted rating (1-5, -1 for invalid): 5
  Actual rating: 4
  Prediction incorrect

Processing text 2/500...
  Text: Don't waste your time with these cables. I bought 2 of
      them. One of them was already broken and the other broke
      after a month or so.

Don't make the same mistake I did: spend a bit more money (even
    just $10 more) and you can get cables that are 1000 times more
     reliable than these. To be honest I'm pretty shocked at how
   many good reviews these are getting. If I had only bought 1
   and it turned out not so good, I might've given them the
   benefit of the doubt, but the fact that both cables were
   complete duds makes it pretty evident that this is just a bad
   product. Bad cables, don't bother with these
  Original output: 1
  Predicted rating (1-5, -1 for invalid): 1
  Actual rating: 1
  Prediction correct

Processing text 3/500...
  Text: I really want to love it...but it's hard to part with $400
      bucks for so little.  I think the price on this should be
      more in the 250 range.  Roland and Mogami...your paying for
      the name.  That being said it is a great little portable amp
      .  If you travel a lot it may be worth the investment. The
      looper is fine, but it only gives you a 40 sec loop.  I like
       my boss looper better as I can store many full size songs.
       It's not as loud (even plugged in) as my little pignose,
      but the anti feedback works very well with my t-5 and
      acoustic taylors.
I really do recommend this amp, I'm just still reeling from
    sticker shock. Nice amp...overpriced
  Original output: 4
  Predicted rating (1-5, -1 for invalid): 4
  Actual rating: 4
  Prediction correct

Processing text 4/500...
  Text: I owned the SN-1 tuner and loved it.  However, the part
      that holds the stem in place broke (note:  Don't carry any
      of these Snark tuners in your pocket!).  I replaced it with
      the SN-8 because it is supposed to be the better model for
      not much more cost. It works okay, but I like the SN-1
```

```
            better.  The display on the SN-1 shows much finer gradations
            of pitch.  The SN-8 has much wider bars and does not
            display a steady reading.  The SN-8 has been harder to use
            than any of my multiple previous tuners. I have used the SN
            -8 (and the SN-1 before it) on an upright bass, guitar,
            mandolin & banjo and it picks up the pitch in any frequency
            range.  Overall, this tuner seems decent for the price.
            Because of the display, I am considering going back to the
            SN-1. As with any tuner, this will get you close, but you
            still need to use your ears for exact fine tuning. Decent
            Tuner For the Price - Like the SN-1 Better
      Original output: 3
      Predicted rating (1-5, -1 for invalid): 3
      Actual rating: 3
      Prediction correct

  Processing text 5/500...
    Text: I've always wanted a Gibson Les Paul, but not being a
            professional I wasn't about to spend thousands for one.  I
            was on the verge of going for a comparable Epiphone model,
            then the OE20TS caught my eye.  Reviews of the favorable
            variety swayed me to go this way, as well as the beautiful
            Tobacco SB finish.  The price didn't hurt either.  The
            guitar arrived packed well. No nicks, scratches, or dents,
            and I received the one I ordered (seems to have been a
            problem for some.)  2 for 2 so far.  The instrument looks
            wonderful.  Nice finish, seems to be put together well, just
            as I had hoped.  Then I played it.  It was obvious that it
            needed quite a bit of adjustment.  There was a ton of fret
            buzz and the intonation was way off.  After making
            adjustments to the truss rod and saddles I was ready to go.
            I have to admit, this thing plays great.  Fantastic tone
            and great sustain.  The action feels sharp and makes it a
            pleasure to play.  The trade off is you get a great guitar
            for a great price, it may just need some setup.  Don't be
            scared off by this if you are a beginner.  There are several
            videos on youtube that show how to make these adjustments,
            and they're really not very difficult.  My guitar came with
            a cable, a hex wrench (for truss adjustment), and a warranty
            card.  Overall I'm very pleased with this purchase. Good
            guitar becomes great with a little TLC.
      Original output: 4
      Predicted rating (1-5, -1 for invalid): 4
      Actual rating: 5
      Prediction incorrect

  Processing text 6/500...
    Text: I am a brand new uke player as of Christmas.  Never played
            anything before in my life. That said, I have been keeping
            my instrument up high for two reasons. 1.) have needed to
            see the strings while I learn. 2.) Because I have had to
            hold my instrument with the inside of my right elbow. It has
            also been difficulty for me to focus on the finger/fret
            movement because I have also had to hold the uke with my
            thumb. It also made my shoulder really g tight. Between the
            shoulders and the thumb and the elbow working so hard, I
            could not relax.  I just got my strap today.  I tried the
            method where the strap goes around the neck and hooks into
            the hole of the uke. That was a bit better tan nothing but
```

```
         the neck of the uke kept flopping around. So I tried the
         method that wraps around the shoulder. That was MUCH better.
           I LOVE the fact that the second way can hold the neck so
         it won't flop around and my thumb does not have to actually
         hold the instrument up.  HOWEVER,  I am a 45 yr adult adult
         and there is simply not enough strap left over to begin to
         wrap around my arm.  I already know that if it was longer I
         could relax my shoulders and drop it down which is probably
         proper form. I don't want to pick up bad habits while I am
         teaching myself.  So I am ordering ANOTHER strap just like
         it so I can lengthen it. I realize I could prob go to a
         fabric store and purchase the same type of strap. But I also
          want the plastic connectors. I intend to modifiy this. It
         is a GREAT idea... but does not work for me as manufactured.
           I hope to come up with a new way that does not go under my
          armpit because that was a little uncomfortable also when I
         moved the strap to the arm end just to see how it would feel
          ( even though then it was WAY too short to reach the neck
         when I did that. This might be alright for a small child, or
          an adult that wants to use is and put it in the hole. But
         if you need the neck of the uke held so it does not flop
         around, this won't work for you...  It is cheap enough  It
         is inexpensive enough that it seems it would easy enough for
          the company to add another $5-10 and add another foot or so
          of strap.  I am going to end up spending that anyway.
         Would be nice to come from the manufacturer ready to use and
          not have to modify it.  I will write another review after I
          see what I  do. That was a bit better tan nothing but the
         neck of the uke kept ...
    Original output: 3
    Predicted rating (1-5, -1 for invalid): 3
    Actual rating: 4
    Prediction incorrect

    ...........
```

**LLM few shot prompting on Toxic**

```
You are a toxicity classifier. I will provide you with a customer
    comment, and you need to determine if the comment is toxic.
Output 1 if the comment is toxic.
Output 0 if the comment is not toxic.

Here are some examples:

Comment: "Wikipedia:Future events

Weighing in on your conversation above; the examples I like to use
    are celestial events and the 2013 inauguration. Lunar and
   solar eclipses are predicted well past the end of this century
   , and the schedule is reliably sourced; there are many pages
   of these, and the possibility of at least one typographical
   error only means they should be checked afainst other
   references. The 2013 inauguation, meanwhile, is years away and
    the inauguree would be speculative, although references to an
    approved budget or scheduling of events would prompt the
   creation of its page, though it may occur before the election,
    and before the inauguree is known, if properly sourced. Wasn'
```

```
    t positive how to link to the policy, but made it the subject
    header. 75.203.136.226"
Toxicity: 0

Comment: "LOL Nawlin, you are so predictable it's funny.

I knew you would spot my edits within a few hours and quickly
    censor them because you are a Commie scum who hates the truth
    ."
Toxicity: 1

Comment: "On the discography it is labeled as both an EP (which it
    is) and a studio album (which it also is).  While it is
    listed as a different color than the other studio albums, it
    is under the list of Studio albums and also included in the
    total count of studio albums."
Toxicity: 0

Comment: "There are two sources already provided for the statement
    that he seems intent on removing, and the issue has been
    discussed in the talk section of the article. The table lists
    National Championships, not NCAA-recognized National
    Championships. Stating that just because the NCAA doesn't yet
    have a sanctioned system in place for recognizing a national
    champion in a sport doesn't make it not a sport. His
    assertions are laughably absurd and he has been reported for
    his violation of 3RR and appears to be using sockpuppets to
    wage an edit war in an article that has already seen too many.
     Please don't indulge this kind of disruptive user."
Toxicity: 0

Comment: "Picking on opinions you do not like is obscene

I followed the discussion. The proposal to userfy all userboxes
    did not succeed, and in the meantime there is nothing ruling
    out political userboxes, not to speak of deleting them from
    the user space. Singling out someone you do not agree with
    like NicholasTurnbull does is just obscene, and as long as you
     do not delete all the userboxes – which I would be perfectly
    ok with – stop picking on mine."
Toxicity: 0

Comment: "Longhair, go take a quick eze.  Spread yr misery
    elsewhere."
Toxicity: 1

Comment: "Relevant to a mature understanding of this topic is
    Keyfabe – a term from the world of professional wrestling, but
     which applies in a wider context.  Individual cases require
    thoughtful judgment, but one thing we should be clear on: not
    everything in tabloids is true.  A fair amount of it is staged
     PR fluff.  Another portion of it is simply bad reporting that
    the stars don't complain about because it is harmless.  There
    are often good reasons to take it all with a grain of salt."
Toxicity: 0

Comment: "a region of the celestial sphere close to the ecliptic."
Toxicity: 0
```

```
Comment: ""
Not true! The actual issue was that you deleted ALL the links I
    added for him. That is the recap. The two other links worked
    fine, but you deleted them as well in your hurry to smash
    everything someone else did. In other words, you are the one
    who jumped to conclusions. Now you're trying to claim you only
     deleted the one that was a problem. Even your edit summary is
     wrong. The man's resignation was presented the same day I was
     adding the links, unknown to me, so it seems the Parliament
    link was being moved from current to former which caused
    pointer problems when I clicked on it. Things happen. It only
    needed to be fixed. I would have thought an Admin would be
    capable of figuring that out. Nor could you figure out how to
    leave a message on my Talk page so I could figure it out. You
    just couldn't wait to smash everything. I only came back to
    the article because I was going through the non-Cabinet people
     on the list. Did you fix that list, once you realized this
    was a former MP? No, you did not. That would have been ""work
    "", requiring ""thought"" and ""effort"". Same as you mis-
    corrected the hat note on the other John Carter MP. Based on
    your arrogance, I thought it likely you were an Admin. What a
    surprise, I was right. And Jimbo wonders why the numbers of
    actual contributors are going straight downhill. I have NO
    intention of continuing to contribute to Wikipedia because you
     obviously would prefer to do everything yourself. You're not
    at all welcoming, helpful, polite, assuming good faith, or
    anything else I was led to believe is part of the Wikipedia
    ethos. So I leave you to it. You didn't even have the grace to
     apologize for deleting the other links, you just tried to
    blame me in this audit trail. Go lecture someone else, because
     I'm out of here, and you can be sure I won't be encouraging
    anyone else to participate in what could have been a good
    project. Your attitude is horrendous, and I'm sure I'm not the
     first one you've chased out of here. Well, keep it up and you
    'll soon have the whole thing to yourself. Enjoy. Go brag to
    your friends that you got rid of another contributor. I'm sure
     you're very proud of yourself.    ""
Toxicity: 0

Comment: "your conclusion (that Ryu isthe next Akuma or whatever)"
Toxicity: 0

Comment: ""

I have not edited that section and I have not yet reviewed it.
    However, it really does not matter what editors think, what
    matters is what experts think and statements made supported by
     reliable references. If you have them, use them. IF you don't
    , then it is just bantering opinions, which is not fruitful. -
    StormRider ""
Toxicity: 0

Comment: ""
What do you mean ""continue""? I did one edit after, which was for
     something completely unrelated.  (talk | )  ""
Toxicity: 0

Comment: ""{| class=""wikitable""
DAMN RIGHT MARK!!! SERVES YOU RIGHT, DONT DO BAD HTINGS!
```

```
|}""
Toxicity: 1

Please output only the integer 0 or 1 corresponding to the
    toxicity, without any other content.
You should use your reasoning skills to make your own decisions
    without consulting your memory or other external sources.

Here is the comment to be classified:

Comment: "Good Work

Thank you for your contribution of File:Coastal City School Bus
    crop.JPG to Green Lantern (film). It is nice to see editors
    such as yourself taking the initiative and going out to
    provide us with free content."
Toxicity:
```

**LLM outputs on Toxic**

```
    Data loaded successfully. Initial shape: (150722, 3)
Shape after removing empty comments: (150722, 3)
Few-shot samples created: 12
Indices used for few-shot: [6435, 13708, 20264, 22988, 34089,
    38490, 39356, 44010, 48934, 64535, 72318, 104250]

Target test set: 500 non-toxic, 20 toxic.

--- DEBUG: Data available for test set (after few-shot exclusion)
    ---
Shape of df_test_candidates: (150710, 3)
Value counts in df_test_candidates['binary_label']:
binary_label
0    143336
1      7374
Name: count, dtype: int64

--- DEBUG: Test set candidate splits ---
Number of toxic candidates for test set: 7374
Number of non-toxic candidates for test set: 143336

--- DEBUG: Test samples BEFORE shuffle ---
Total items in test_samples (before shuffle): 520
First 25 labels before shuffle: [1, 1, 1, 1, 1, 1, 1, 1, 1, 1, 1,
    1, 1, 1, 1, 1, 1, 1, 1, 1, 0, 0, 0, 0, 0]
Last 25 labels before shuffle: [0, 0, 0, 0, 0, 0, 0, 0, 0, 0, 0,
    0, 0, 0, 0, 0, 0, 0, 0, 0, 0, 0, 0, 0, 0]
Counts before shuffle: 0s=500, 1s=20

Test set size (after shuffle): 520
Test set composition (true_labels overall): 20 toxic (label 1),
    500 non-toxic (label 0).
First 50 true_labels after shuffle: [0, 0, 0, 0, 0, 0, 0, 0, 0, 0,
    0, 0, 0, 0, 0, 0, 0, 1, 0, 0, 0, 0, 0, 0, 0, 0, 0, 0, 0, 0,
    0, 0, 0, 0, 0, 0, 0, 0, 0, 0, 0, 0, 1, 0, 0, 0, 0, 0]
Processing text 1/520...

  Comment: Notability of Ambridge country club
```

```
A tag has been placed on Ambridge country club, requesting that
   ...
  Original LLM output: 0
  Predicted label (0/1, -1 for invalid): 0
  Actual label: 0

Processing text 2/520...
  Comment: "

Pakistani Language (Urdu)
Urdu in Persian (Parsi) means camp and ``Urdu'' language was the
   langua...
  Original LLM output: 1
  Predicted label (0/1, -1 for invalid): 1
  Actual label: 0

Processing text 3/520...
  Comment: ":::::::What I find a bit strange about the various
     guidelines and FAQ pages about categories is tha...
  Original LLM output: 0
  Predicted label (0/1, -1 for invalid): 0
  Actual label: 0

Processing text 4/520...
  Comment: Re:What is it with me and categories (very polite title
     )
He's a Dark Lord. That's the same thing....
  Original LLM output: 0
  Predicted label (0/1, -1 for invalid): 0
  Actual label: 0

Processing text 5/520...
  Comment: Plagerism
This entry is almost entirely taken from [].  It is copyrighted.
     Please fix the immediat...
  Original LLM output: 0
  Predicted label (0/1, -1 for invalid): 0
  Actual label: 0

Processing text 6/520...
  Comment: "

 ""The use of the bombs""?

Could the subsection titled ""The use of the bombs"" be renamed ""
     Bo...
  Original LLM output: 0
  Predicted label (0/1, -1 for invalid): 0
  Actual label: 0

Processing text 7/520...
  Comment: "

Hi Smokefoot: Thanks for your comments. It became apparent that I
     could do further edits only aft...
  Original LLM output: 0
  Predicted label (0/1, -1 for invalid): 0
  Actual label: 0
```

```
Processing text 8/520...
  Comment: Photographs
A couple of photographs, at least, exist of Dilwar. I don't know
   the copyright position ...
  Original LLM output: 0
  Predicted label (0/1, -1 for invalid): 0
  Actual label: 0

Processing text 9/520...
  Comment: 1. search news about steam valve 2. add to steam
      article whilst ignoring usefulness of content.

NO....
  Original LLM output: 0
  Predicted label (0/1, -1 for invalid): 0
  Actual label: 0

Processing text 10/520...
  Comment: "

 TVMediaInsights

Recent discoveries show that people who work for this website tend
    to be posti...
  Original LLM output: 0
  Predicted label (0/1, -1 for invalid): 0
  Actual label: 0

...........
```

- **Writing aid and polishing:** LLMs were used to assist in improving grammar, clarity, and style. The substantive content, ideas, and technical contributions remain the authors' own.

- **Retrieval and discovery:** LLMs were employed to support literature search and discovery (e.g., identifying related work). All cited references were verified by the authors.

