# OpenReview forum: "Structuring Semantic Embeddings for Principle Evaluation: A Kernel-Guided Contrastive Learning Approach"
_ICLR.cc/2026/Conference — ICLR 2026 Conference Withdrawn Submission_

### Official Review · Reviewer_Ehqo · 2025-10-29

**Soundness:** 2
**Presentation:** 2
**Contribution:** 2
**Rating:** 2
**Confidence:** 3

**Summary:**

This paper concerns on evaluating whether synthetic text can adhere to human-defined principles. This is a practical problem to align AI and human. The authors seem to transform this into a classification problem in which each principle is considered as a class. Then a contrastive learning framework is proposed to learn text embeddings. The goodness of those embeddings is evaluated using downstream tasks, including emotion classification and toxicity detection. Finally, the authors did experiments on GoEmotions, Amazon reviews, and toxic comments datasets to evaluate the learned embeddings, in comparison with the raw text.

**Strengths:**

- The main problem of interest is practical and relevant.
- The trained embeddings seem to be better than the raw representation.

**Weaknesses:**

- Although focusing on principle adherence, the authors do not formally discuss about principles. This leads to an unclear research problem in this paper.
- There seems to be no explicit kernel in their proposed framework, despite repeated mentions about "kernel-based" in the paper.
- No baseline is used for comparison. The main framework focuses on learning representation for the text. So, the authors should take some existing methods that can learn text embeddings into comparison.
- The current manuscript lacks lot of details. For example, AdaptedInfoNCE is used without definition.
- The experiments focus on some supervised problems, but not the main concern of the paper about human-defined principles. This means, the experimental part may not support well the claim of this paper.

**Questions:**

Can you define principles explicitly?

---

### Official Review · Reviewer_yaX4 · 2025-11-01

**Soundness:** 2
**Presentation:** 2
**Contribution:** 2
**Rating:** 2
**Confidence:** 4

**Summary:**

The paper introduces a kernel-guided contrastive learning framework for evaluating how well text aligns with abstract human principles such as fairness or honesty. Conventional text embeddings often mix principle-related signals with general semantic information, making evaluation unreliable. To address this, the authors project embeddings into a structured subspace where each principle is represented by a learnable prototype vector and the surrounding geometry is shaped by an offset penalty that enforces clear separation while preserving contextual nuance. Through joint optimization of contrastive and regularization losses, the model learns embeddings that better capture the essence of each principle without discarding general meaning. Experiments across emotion, sentiment, and toxicity datasets show that these structured embeddings yield more accurate classification and regression results than raw embeddings or few-shot large language models, indicating that purpose-built representations can enhance reliability in principle evaluation.

**Strengths:**

- The paper correctly identifies that post-hoc value or principle evaluation is underexplored compared to controllable generation methods.
- The approach reframes alignment evaluation as a representation optimization problem, which is a useful perspective that could inspire modular or plug-in evaluators.
- The experiments report cross-validation and multiple downstream models, providing some statistical grounding for the claims.

**Weaknesses:**

### w1. Conceptual Ambiguity: “Principle Evaluation” vs. “Value Alignment Evaluation”

The paper frames its goal as “principle evaluation” but positions it as a step toward “value alignment evaluation.” The distinction is unclear:

- Value alignment typically concerns consistency between model and human values.
- Principle evaluation here appears to test adherence to explicit, human-defined rules.

If these are equivalent, standard alignment terminology should be used; if not, the boundary between the two should be clarified and the focus on “principles” justified. This ambiguity weakens the conceptual framing.

### w2. Limited Experimental Coverage

Experiments are limited to one embedding model (Jina v3) and three proxy datasets.

- The tasks (emotion, sentiment, toxicity) evaluate classification separability but not “principle adherence” or “value alignment.”
- Comparison with few-shot LLMs is not directly relevant, as prompting-based generation differs fundamentally from embedding-level evaluation.

Without alignment-specific benchmarks, it is difficult to assess whether the proposed framework advances value-alignment evaluation.

### w3. Missing Alignment Evaluation Baselines

If the goal is value alignment evaluation, the paper should compare against existing alignment evaluators or benchmarks. Instead, it relies on few-shot LLMs and proxy tasks that assess classification rather than alignment. This makes it unclear whether the method improves on standard classification. If the task differs from conventional alignment evaluation, this distinction and its rationale should be explicitly stated.

### w4. Use of “Kernel” Terminology

The paper uses “kernel” to describe learnable prototype vectors acting as embedding anchors. However, these are not kernel functions in the classical RKHS sense, and no kernelized similarity is applied. This unconventional terminology may confuse readers familiar with kernel methods. Unless a formal connection is provided, “prototype vectors” or “anchors” would be clearer.

### w5. Limited Qualitative and Interpretability Analysis

While t-SNE visualizations, kernel–principle correlations, and offset analyses illustrate geometric structure, they remain low-level and do not reveal how learned “principles” map to human-interpretable semantics. More qualitative examples or probing analyses (e.g., principle-specific attribution or activation studies) would strengthen the interpretability claims.

**Questions:**

Please refer to the Weaknesses.

---

### Official Review · Reviewer_StWr · 2025-11-03

**Soundness:** 2
**Presentation:** 2
**Contribution:** 1
**Rating:** 2
**Confidence:** 4

**Summary:**

This paper introduces a kernel-guided contrastive learning framework that uses learnable prototype kernels and a novel offset penalty to restructure fixed embeddings, forcing disentanglement of principle-specific features for post-hoc evaluation

**Strengths:**

The optimized embeddings consistently yield statistically significant performance improvements compared to raw features and demonstrably outperform few-shot LLMs on structured proxy tasks.

The resulting structured, low-dimensional subspace provides a reusable intermediate representation that simplifies downstream modelling and improves computational efficiency.

**Weaknesses:**

TL;DR: while the work offers some performance improvements, the proposed method is largely an amalgamation of known techniques without significant novelty so I would not consider it appropriate for ICLR and suggest submitting to an alternative venue.

*  Excessive Loss Complexity: The method relies on a highly composite loss function $L_{\text{total}}$ combining five distinct, weighted terms (Contrastive, Offset, Classification, Orthogonality, Magnitude). This fragility implies that no single mechanism is robust, mandating extensive hyperparameter tuning (e.g., numerous $\lambda$ weights, $\tau$, $\delta_{\text{intra}}$, $\delta_{\text{inter}}$).
*  Derivativity of Core Components: The introduction of "learnable prototype kernels" is an architectural modification of existing Prototypical Contrastive Learning (PCL), and the "novel Offset Loss" is fundamentally a configuration of standard metric learning distance margin constraints applied via Euclidean distance penalties.
* Verification Gap in Problem Scope: The framework is motivated by the "critical challenge" of evaluating abstract principles (e.g., fairness, honesty, safety). However, empirical validation is confined to more measurable, structured proxy tasks, namely fine-grained emotion classification (GoEmotions), ordinal star ratings (Amazon Reviews), and binary toxicity detection. This validation scope avoids the subtle, context-dependent, and inherently subjective principles that prompted the research.
* Non-Rigorous Baseline Comparison: The highlighted superior performance against few-shot Large Language Models (LLMs) lacks rigour. The comparison contrasts a specialized, supervised transformation network against generalist LLMs restricted to simple prompting. A meaningful assessment requires benchmarking against state-of-the-art specialized methods in supervised contrastive learning or deep metric learning.

**Questions:**

see Weaknesses

---

### Official Review · Reviewer_nLQu · 2025-11-03

**Soundness:** 1
**Presentation:** 2
**Contribution:** 1
**Rating:** 2
**Confidence:** 4

**Summary:**

This paper proposes a kernel-guided contrastive learning framework to improve text embeddings for "principle evaluation" - essentially classifying text according to various attributes like emotions, toxicity, or ratings. The authors argue that standard embeddings entangle principle-specific signals with general semantic content, making evaluation difficult. Their solution involves learning prototype kernels for each "principle" and using a complex multi-component loss function to structure a 64-dimensional embedding space. They test on three datasets: GoEmotions (emotion classification), Amazon Reviews (rating prediction), and toxic comment detection.

**Strengths:**

- The paper includes proper statistical reporting with cross-validation and provides extensive implementation details in the appendix.
- The optimized embeddings do show performance improvements over raw embeddings across the tested scenarios, even if the gains are sometimes modest.
- Thorough ablation study: The authors validate the contribution of different loss components.

**Weaknesses:**

- I do not understand the core motivation of this work: The claim that post-hoc evaluation is "less explored" compared to generation-time control is not correct. Evaluation metrics and classifiers are tools that drive AI safety development. The motivation for why embeddings are the preferred approach "at scale" over specialized classifiers is not clear.
- The paper conflates completely different tasks (emotion classification, toxicity detection, rating prediction) under the vague umbrella of "principle evaluation." These are distinct problems different characteristics.
- Overcomplicated solution: The framework involves different loss components with numerous hyperparameters. For what amounts to learning better representations, this seems unnecessarily complex compared to standard approaches.
- Unclear technical contributions: The "learnable prototype kernels" are essentially learned class centroids, which isn't novel.

The paper reads as if it's searching a problem, applying complex machinery to what seems to be standard classification tasks with different embedding pre-processing.

**Questions:**

The paper is confusing about whether they're learning principle-specific dimensions or just mapping to 64 dimensions. The relationship between the "structured subspace" and the actual 64-dimensional output is unclear.

---

### Note · Authors · 2025-12-13

I have read and agree with the venue's withdrawal policy on behalf of myself and my co-authors.